# Monomial Matrix Group Equivariant Neural Functional Networks

**Viet-Hoang Tran**[*]
Department of Mathematics
National University of Singapore
`hoang.tranviet@u.nus.edu`

**Thieu N. Vo**[*]
Department of Mathematics
National University of Singapore
`thieuvo@nus.edu.sg`

**Tho Tran-Huu**
Department of Mathematics
National University of Singapore
`thotranhuu@u.nus.edu.vn`

**An T. Nguyen**
FPT Software AI Center
`annt68@fpt.com`

**Tan M. Nguyen**
Department of Mathematics
National University of Singapore
`tanmn@nus.edu.sg`

## Abstract

Neural functional networks (NFNs) have recently gained significant attention due to their diverse applications, ranging from predicting network generalization and network editing to classifying implicit neural representation. Previous NFN designs often depend on permutation symmetries in neural networks' weights, which traditionally arise from the unordered arrangement of neurons in hidden layers. However, these designs do not take into account the weight scaling symmetries of $\mathrm{ReLU}$ networks, and the weight sign flipping symmetries of $\mathrm{sin}$ or $\mathrm{Tanh}$ networks. In this paper, we extend the study of the group action on the network weights from the group of permutation matrices to the group of monomial matrices by incorporating scaling/sign-flipping symmetries. Particularly, we encode these scaling/sign-flipping symmetries by designing our corresponding equivariant and invariant layers. We name our new family of NFNs the Monomial Matrix Group Equivariant Neural Functional Networks (Monomial-NFN). Because of the expansion of the symmetries, Monomial-NFN has much fewer independent trainable parameters compared to the baseline NFNs in the literature, thus enhancing the model's efficiency. Moreover, for fully connected and convolutional neural networks, we theoretically prove that all groups that leave these networks invariant while acting on their weight spaces are some subgroups of the monomial matrix group. We provide empirical evidences to demonstrate the advantages of our model over existing baselines, achieving competitive performance and efficiency. The code is publicly available at https://github.com/MathematicalAI-NUS/Monomial-NFN.

## 1 Introduction

Deep neural networks (DNNs) have become highly versatile modeling tools, finding applications across a broad spectrum of fields such as Natural Language Processing [15, 29, 51, 66], Computer

---

[*]Equal contribution. Please correspond to `thieuvo@nus.edu.sg`.

38th Conference on Neural Information Processing Systems (NeurIPS 2024).

Vision [27, 37, 61], and the Natural Sciences [31, 49]. There has been growing interest in developing specialized neural networks to process the weights, gradients, or sparsity masks of DNNs as data. These specialized neural networks are called neural functional networks (NFNs) [71]. NFNs have found diverse applications, ranging from predicting network generalization and network editing to classifying implicit neural representations. For instance, NFNs have been employed to create learnable optimizers for neural network training [5, 11, 42, 52], extract information from implicit neural representations of data [43, 52, 60], perform corrective editing of network weights [13, 44, 56], evaluate policies [26], and conduct Bayesian inference using networks as evidence [59].

Developing NFNs is inherently challenging due to their high-dimensional nature. Some early methods to address this challenge assume a restricted training process that effectively reduced the weight space [9, 19, 41]. More recent efforts have focused on building permutation equivariant NFNs that can process neural network weights without such restrictions [35, 45, 71, 72]. These works construct NFNs that are equivariant to permutations of weights, corresponded to the rearrangement of neurons in hidden layers. Such permutations, known as neuron permutation symmetries, preserve the network's behavior. However, these approaches often overlook other significant symmetries in weight spaces [12, 24]. Notable examples are weight scaling transformations for $\mathrm{ReLU}$ networks [7, 12, 46] and sign flipping transformations for $\sin$ and $\tanh$ networks [14, 22, 38]. Consequently, two weight spaces of a $\mathrm{ReLU}$ networks, that differ by a scaling transformation, two weight spaces of a $\sin$, or $\tanh$ networks that differ by a sign flipping transformation, can produce different results when processed by existing permutation equivariant NFNs, despite representing the same functions. This highlights a fundamental limitation of the current permutation equivariant NFNs.

**Contribution.** In this paper, we extend the study of symmetries in weight spaces of Fully Connected Neural Networks (FCNNs) and Convolution Neural Networks (CNNs) by formally establishing a group of symmetries that includes both neuron permutations and scaling/sign-flipping transformations. These symmetries are represented by monomial matrices, which share the nonzero pattern of permutation matrices but allow nonzero entries to be any value rather than just 1. We then introduce a novel family of NFNs that are equivariant to groups of monomial matrices, thus incorporating both permutation and scaling/sign-flipping symmetries into the NFN design. We name this new family Monomial Matrix Group Equivariant Neural Functional Networks (Monomial-NFNs). Due to the expanded set of symmetries, Monomial-NFN requires significantly fewer independent trainable parameters compared to baseline NFNs, enhancing the model's efficiency. By incorporating equivariance to neuron permutations and weight scaling/sign-flipping, our NFNs demonstrate competitive generalization performance compared to existing models. Our contribution is three-fold:

1. We formally describe a group of monomial matrices satisfying the condition that the transformation of weight spaces of FCNNs and CNNs using these group elements does not change the function defined by the networks. For $\mathrm{ReLU}$ networks, this group covers permutation and scaling symmetries of the weight spaces, while for $\sin$ or $\tanh$ networks, this group covers permutation and sign-flipping symmetries. The group is proved to be maximal in certain cases.

2. We design Monomial-NFNs, the first family of NFNs that incorporate scaling and sign-flipping symmetries of weight spaces as far as we are aware. The main building blocks of Monomial-NFNs are the equivariant and invariant linear layers for processing weight spaces.

3. We show that the number of parameters in our equivariant linear layer is much lower than in recent permutation equivariant NFNs. In particular, our method is linear in the number of layers and dimensions of weights and biases, compared to quadratic as in [71]. This demonstrates that Monomial-NFNs have the ability to process weight spaces of large-scale networks.

We evaluate Monomial-NFNs on three tasks: predicting CNN generalization from weights using Small CNN Zoo [64], weight space style editing, and classifying INRs using INRs data [71]. Experimental results show that our model achieves competitive performance and efficiency compared to existing baselines.

**Organization.** We structure this paper as follows: After summarizing some related work in Section 2, we recall the notions of monomial matrix group and describe their maximal subgroups preserved by some nonlinear activations in Section 3. In Section 4, we formalize the general weight space of FCNNs and CNNs, then discuss the symmetries of these weight spaces using the monomial matrices. In Section 5, we construct monomial matrix group equivariant and invariant layers, which are building blocks for our Monomial-NFNs. In Section 6, we present our experimental results to justify the advantages of Monomial-NFNs over the existing permutation equivariant NFN baselines. The paper ends with concluding remarks. More experimental details are provided in the Appendix.

## 2 Related Work

**Symmetries of Weight Spaces.** The challenge of identifying the symmetries in the weight spaces of neural networks, or equivalently, determining the functional equivalence of neural networks, is a well-explored area in academic research [3, 10, 16, 23, 47]. This problem was initially posed by Hecht-Nielsen in [28]. Results for various types of networks have been established as in [1, 2, 12, 14, 22, 38, 63].

**Neural Functional Networks.** Recent research has focused on learning representations for trained classifiers to predict their generalization performance and other insights into neural networks [8, 20, 64, 54, 53, 55]. In particular, low-dimensional encodings for Implicit Neural Representations (INRs) have been developed for downstream tasks [18, 41]. Other studies have encoded and decoded neural network parameters mainly for reconstruction and generation purposes [6, 21, 34, 48].

**Equivariant Neural Functional Networks.** Permutations and scaling, for ReLU networks, as well as sign-flipping, for sine or tanh networks, symmetries, are fundamental symmetries of weight networks. Permutation-equivariant NFNs are successfully built in [4, 35, 40, 45, 70, 71, 72]. In particular, the authors in [35, 40] carefully construct computational graphs representing the input neural networks' parameters and process the graphs using graph neural networks. In [4], neural network parameters are efficiently encoded by carefully choosing appropriate set-to-set and set-to-vector functions. The authors in [70] view network parameters as a special case of a collection of tensors and then construct maximally expressive equivariant linear layers for processing any collection of tensors given a description of their permutation symmetries. These methods are applicable to several types of networks, including those with branches or transformers. However, these models were not necessarily equivariant to scaling nor sign-flipping transformations, which are important symmetries of the input neural networks.

Our method makes the first step toward incorporating both permutation and non-permutation symmetries into NFNs. In particular, the model proposed in our paper is equivariant to permutations and scaling, for ReLU networks, or sign-flipping, for sine and tanh networks. This leads to a significant reduction in the number of parameters, a property that is particularly useful for large neural networks in modern deep learning, while achieving comparable or better results than those in the literature. The authors in [32, 67] have also developed NFNs that incorporates scaling symmetries.

## 3 Monomial Matrices Perserved by a Nonlinear Activation

Given two sets $X, Y$, and a group $G$ acts on them, a function $\phi\colon X \to Y$ is called $G$-equivariant if $\phi(g \cdot x) = g \cdot \phi(x)$ for all $x \in X$ and $g \in G$. If $G$ acts trivially on $Y$, then we say $\phi$ is $G$-invariant. In this paper, we consider NFNs which are equivariant with respect to certain symmetries of deep weight spaces. These symmetries will be represented by monomial matrices. In Subsection 3.1, we recall the notion of monomial matrices, as well as their actions on space of matrices. We then formalize the maximal group of matrices preserved by the activations $\mathrm{ReLU}$, $\sin$ and $\mathrm{Tanh}$ in Subsection 3.2.

### 3.1 Monomial Matrices and Monomial Matrix Group Actions

All matrices considered in this paper have real entries and $n$ is a positive integer.

**Definition 3.1** (See [50, page 46])**.** A matrix of size $n \times n$ is called a *monomial matrix* (or *generalized permutation matrix*) if it has exactly one non-zero entry in each row and each column, and zeros elsewhere. We will denote by $\mathcal{G}_n$ the set of such all matrices.

*Permutation matrices* and *invertible diagonal matrices* are special cases of monomial matrices. In particular, a permutation matrix is a monomial matrix in which the non-zero entries are all equal to $1$. In case the nonzero entries of a monomial matrix are in the diagonal, it becomes an invertible diagonal matrix. We will denote by $\mathcal{P}_n$ and $\Delta_n$ the sets of permutation matrices and invertible diagonal matrices of size $n \times n$, respectively. It is well-known that the groups $\mathcal{G}_n$, $\mathcal{P}_n$, and $\Delta_n$ are subgroups of the general linear group $\mathrm{GL}(n)$.

Permutation matrix group $\mathcal{P}_n$ is a representation of the permutation group $S_n$, which is the group of all permutations of the set $\{1, 2, \ldots, n\}$ with group operator as the composition. Indeed, for each permutation $\pi \in S_n$, we denote by $P_\pi$ the square matrix obtained by permuting $n$ columns of the identity matrix $I_n$ by $\pi$. We call $P_\pi$ the *permutation matrix* corresponding to $\pi$. The correspondence $\pi \mapsto P_\pi$ defines a group homomorphism $\rho\colon S_n \to \mathrm{GL}(n)$ with the image $\mathcal{P}_n = \rho(S_n)$.

Each monomial matrix in $\mathcal{G}_n$ is a product of an invertible diagonal matrix in $\Delta_n$ and a permutation matrix in $\mathcal{P}_n$, i.e.

$$\mathcal{G}_n = \{DP \; : \; D \in \Delta_n \text{ and } P \in \mathcal{P}_n\}. \tag{1}$$

In general, we have $PD \neq DP$. However, for $D = \operatorname{diag}(d_1, d_2, \ldots, d_n)$ and $P = P_\pi$, we have $PD = (PDP^{-1})P$ which is again a product of the invertible diagonal matrix

$$PDP^{-1} = \operatorname{diag}(d_{\pi^{-1}(1)}, d_{\pi^{-1}(2)}, \ldots, d_{\pi^{-1}(n)}) \tag{2}$$

and the permutation matrix $P$. As an implication of Eq. (2), there is a group homomorphism $\varphi \colon \mathcal{P}_n \to \operatorname{Aut}(\Delta_n)$, defined by the conjugation, i.e. $\varphi(P)(D) = PDP^{-1}$ for all $P \in \mathcal{P}_n$ and $D \in \Delta_n$. The map $\varphi$ defines the group $\mathcal{G}_n$ as the semidirect product $\mathcal{G}_n = \Delta_n \rtimes_\varphi \mathcal{P}_n$ (see [17]). For convenience, we sometimes denote element $DP$ of $\mathcal{G}_n$ as a pair $(D, P)$.

The groups $\mathcal{G}_n$, $\mathcal{P}_n$ and $\Delta_n$ act on the left and the right of $\mathbb{R}^n$ and $\mathbb{R}^{n \times m}$ in a canonical way (by matrix-vector or matrix-matrix multiplications). More precisely, we have:

**Proposition 3.2.** *Let $\mathbf{x} \in \mathbb{R}^n$ and $A = (A_{ij}) \in \mathbb{R}^{n \times m}$. Then for $D = \operatorname{diag}(d_1, \ldots, d_n) \in \Delta_n$, $\overline{D} = \operatorname{diag}(\overline{d}_1, \ldots, \overline{d}_m) \in \Delta_m$, $P_\pi \in \mathcal{P}_n$, and $P_\sigma \in \mathcal{P}_m$, we have:*

$$P_\pi \cdot \mathbf{x} = (x_{\pi^{-1}(1)}, x_{\pi^{-1}(2)}, \ldots, x_{\pi^{-1}(n)})^\top,$$

$$D \cdot \mathbf{x} = (d_1 \cdot x_1, d_2 \cdot x_2, \ldots, d_n \cdot x_n)^\top,$$

$$\left(D \cdot P_\pi \cdot A \cdot P_\sigma \cdot \overline{D}\right)_{ij} = d_i \cdot A_{\pi^{-1}(i)\sigma(j)} \cdot \overline{d}_j,$$

$$\left(D \cdot P_\pi \cdot A \cdot (\overline{D} \cdot P_\sigma)^{-1}\right)_{ij} = d_i \cdot A_{\pi^{-1}(i)\sigma^{-1}(j)} \cdot \overline{d}_j^{-1}.$$

The above proposition can be verified by a direct computation, and is used in subsequent sections.

## 3.2 Monomial Matrices Preserved by a Nonlinear Activation

We characterize the maximal matrix groups preserved by the activations $\sigma = \operatorname{ReLU}, \sin$ or $\tanh$. Here, $\operatorname{ReLU}$ is the rectified linear unit activation function which has been used in most of modern neural networks, $\sin$ is the sine function which is often used as an activation function in implicit neural representations [57], and $\tanh$ is the hyperbolic tangent activation function. Different variants of the results in this subsection can also be found in [24, 68]. We refine them using the terms of monomial matrices and state explicitly here for the completeness of the paper.

**Definition 3.3.** A matrix $A \in \operatorname{GL}(n)$ is said to be *preserved by an activation $\sigma$* if and only if $\sigma(A \cdot \mathbf{x}) = A \cdot \sigma(\mathbf{x})$ for all $\mathbf{x} \in \mathbb{R}^n$.

We adopt the term *matrix group preserved by an activation* from [68]. This term is then referred to as the *intertwiner group of an activation* in [24].

**Proposition 3.4.** *For every matrix $A \in \operatorname{GL}(n)$, we have:*

(i) *$A$ is preserved by the activation $\operatorname{ReLU}$ if and only if $A \in \mathcal{G}_n^{>0}$. Here, $\mathcal{G}_n^{>0}$ is the subgroup of $\mathcal{G}_n$ containing only monomial matrices whose nonzero entries are positive numbers.*

(ii) *$A$ is preserved by the activation $\sigma = \sin$ or $\tanh$ if and only if $A \in \mathcal{G}_n^{\pm 1}$. Here, $\mathcal{G}_n^{\pm 1}$ is the subgroup of $\mathcal{G}_n$ containing only monomial matrices whose nonzero entries are $\pm 1$.*

A detailed proof of Proposition 3.4 can be found in Appendix C.1. As a consequence of the above theorem, $\mathcal{G}_n^{>0}$ (respectively, $\mathcal{G}_n^{\pm 1}$) is the maximal matrix subgroup of the general linear group $\operatorname{GL}(n)$ that is preserved by the activation $\operatorname{ReLU}$ (respectively, $\sin$ and $\tanh$).

**Remark 3.5.** *Intuitively, $\mathcal{G}_n^{>0}$ is generated by permuting and positive scaling the coordinates of vectors in $\mathbb{R}^n$, while $\mathcal{G}_n^{\pm 1}$ is generated by permuting and sign flipping. Formally, these groups can be written as the semidirect products:*

$$\mathcal{G}_n^{>0} = \Delta_n^{>0} \rtimes_\varphi \mathcal{P}_n, \quad \text{and} \quad \mathcal{G}_n^{\pm 1} = \Delta_n^{\pm 1} \rtimes_\varphi \mathcal{P}_n,$$

*where*

$$\Delta_n^{>0} = \{D = \operatorname{diag}(d_1, \ldots, d_n) \; : \; d_i > 0\}, \text{ and} \tag{3}$$

$$\Delta_n^{\pm 1} = \{D = \operatorname{diag}(d_1, \ldots, d_n) \; : \; d_i \in \{-1, 1\}\} \tag{4}$$

*are two subgroups of $\Delta_n$.*

# 4 Weight Spaces and Monomial Matrix Group Actions on Weight Spaces

In this section, we formulate the general structure of the weight spaces of FCNNs and CNNs. We then determine the group action on these weight spaces using monomial matrices. The activation function $\sigma$ using on the considered FCNNs and CNNs are assumed to be $\mathrm{ReLU}$ or $\sin$ or $\mathrm{Tanh}$.

## 4.1 Weight Spaces of FCNNs and CNNs

**Weight Spaces of FCNNs.** Consider an FCNN with $L$ layers, $n_i$ neurons at the $i$-th layer, and $n_0$ and $n_L$ be the input and output dimensions, together with the activation $\sigma$, as follows:

$$f(\mathbf{x} \; ; \; U, \sigma) = W^{(L)} \cdot \sigma \left( \ldots \sigma \left( W^{(2)} \cdot \sigma \left( W^{(1)} \cdot \mathbf{x} + b^{(1)} \right) + b^{(2)} \right) \ldots \right) + b^{(L)}. \tag{5}$$

Here, $U = (W, b)$ is the parameters with the weights $W = \{W^{(i)} \in \mathbb{R}^{n_i \times n_{i-1}}\}_{i=1}^L$ and the biases $b = \{b^{(i)} \in \mathbb{R}^{n_i \times 1}\}_{i=1}^L$. The pair $U = (W, b)$ belongs to the weight space $\mathcal{U} = \mathcal{W} \times \mathcal{B}$, where:

$$\mathcal{W} = \mathbb{R}^{n_L \times n_{L-1}} \times \ldots \times \mathbb{R}^{n_2 \times n_1} \times \mathbb{R}^{n_1 \times n_0}, \tag{6}$$

$$\mathcal{B} = \mathbb{R}^{n_L \times 1} \times \ldots \times \mathbb{R}^{n_2 \times 1} \times \mathbb{R}^{n_1 \times 1}. \tag{7}$$

**Weight Spaces of CNNs.** Consider a CNN with $L$ convolutional layers, ending with an average pooling layer then fully connected layers, together with activation $\sigma$. Let $n_i$ and $w_i$ be the number of channels and the size of the convolutional kernel at the $i^{\text{th}}$ convolutional layer. We will only take account of the $L$ convolutional layers, since the weight space of the fully connected layers are already considered above, and the pooling layer has no learnable parameters:

$$f(\mathbf{x} \; ; \; U, \sigma) = \sigma \left( W^{(L)} * \sigma \left( \ldots \sigma \left( W^{(2)} * \sigma \left( W^{(1)} * \mathbf{x} + b^{(1)} \right) + b^{(2)} \right) \ldots \right) + b^{(L)} \right) \tag{8}$$

Here, $U = (W, b)$ is the learnable parameters with the weights $W = \{W^{(i)} \in \mathbb{R}^{w_i \times n_i \times n_{i-1}}\}_{i=1}^L$ and the biases $b = \{b^{(i)} \in \mathbb{R}^{1 \times n_i \times 1}\}_{i=1}^L$. The convolutional operator $*$ is defined depending on the purpose of the model, and adding $b$ means adding $b_j^{(i)}$ to all entries of $j^-$th channel at $i^{\text{th}}$ layer. The pair $U = (W, b)$ belongs to the weight space $\mathcal{U} = \mathcal{W} \times \mathcal{B}$, where:

$$\mathcal{W} = \mathbb{R}^{w_L \times n_L \times n_{L-1}} \times \ldots \times \mathbb{R}^{w_2 \times n_2 \times n_1} \times \mathbb{R}^{w_1 \times n_1 \times n_0}, \tag{9}$$

$$\mathcal{B} = \mathbb{R}^{1 \times n_L \times 1} \times \ldots \times \mathbb{R}^{1 \times n_2 \times 1} \times \mathbb{R}^{1 \times n_1 \times 1}. \tag{10}$$

**Remark 4.1.** *See in Appendix. C.2 for concrete descriptions of weight spaces of FCNNs and CNNs.*

## 4.2 Monomial Matrix Group Action on Weight Spaces

The **weight space** $\mathcal{U}$ of an FCNN or CNN with $L$ layers and $n_i$ channels at $i^{\text{th}}$ layer has the general form $\mathcal{U} = \mathcal{W} \times \mathcal{B}$, where:

$$\mathcal{W} = \mathbb{R}^{w_L \times n_L \times n_{L-1}} \times \ldots \times \mathbb{R}^{w_2 \times n_2 \times n_1} \times \mathbb{R}^{w_1 \times n_1 \times n_0}, \tag{11}$$

$$\mathcal{B} = \mathbb{R}^{b_L \times n_L \times 1} \times \ldots \times \mathbb{R}^{b_2 \times n_2 \times 1} \times \mathbb{R}^{b_1 \times n_1 \times 1}. \tag{12}$$

Here, $n_i$ is the number of channels at the $i^{\text{th}}$ layer, in particular, $n_0$ and $n_L$ are the number of channels of input and output; $w_i$ is the dimension of weights and $b_i$ is the dimension of the biases in each channel at the $i$-th layer. The dimension of the weight space $\mathcal{U}$ is:

$$\dim \mathcal{U} = \sum_{i=1}^L \left( w_i \times n_i \times n_{i-1} + b_i \times n_i \times 1 \right). \tag{13}$$

**Notation.** When working with weight matrices in $\mathcal{W}$, the space $\mathbb{R}^{w_i \times n_i \times n_{i-1}} = (\mathbb{R}^{w_i})^{n_i \times n_{i-1}}$ at the $i^{\text{th}}$ layer will be considered as the space of $n_i \times n_{i-1}$ matrices, whose entries are real vectors in $\mathbb{R}^{w_i}$. s In particular, the symbol $W^{(i)}$ denotes a matrix in $\mathbb{R}^{w_i \times n_i \times n_{i-1}} = (\mathbb{R}^{w_i})^{n_i \times n_{i-1}}$, while $W_{jk}^{(i)} \in \mathbb{R}^{w_i}$ denotes the entry at row $j$ and column $k$ of $W^{(i)}$. Similarly, the notion $b^{(i)}$ denotes a bias column vector in $\mathbb{R}^{b_i \times n_i \times 1} = (\mathbb{R}^{b_i})^{n_i \times 1}$, while $b_j^{(i)} \in \mathbb{R}^{b_i}$ denotes the entry at row $j$ of $b^{(i)}$.

To define the **group action of** $\mathcal{U}$ using monomial matrices, denote $\mathcal{G}_{\mathcal{U}}$ as the group:

$$\mathcal{G}_{\mathcal{U}} \coloneqq \mathcal{G}_{n_L} \times \ldots \times \mathcal{G}_{n_1} \times \mathcal{G}_{n_0}.$$

Ideally, each monomial matrix group $\mathcal{G}_{n_i}$ will act on the weights and the biases at the $i^{\text{th}}$ layer of the network. Each element of $\mathcal{G}_{\mathcal{U}}$ will be of the form $g = \left( g^{(L)}, \ldots, g^{(0)} \right)$, where:

$$g^{(i)} = D^{(i)} \cdot P_{\pi_i} = \mathrm{diag} \left( d_1^{(i)}, \ldots, d_{n_i}^{(i)} \right) \cdot P_{\pi_i} \in \mathcal{G}_{n_i} \tag{14}$$

for some invertible diagonal matrix $D^{(i)}$ and permutation matrix $P_{\pi_i}$. The action of $\mathcal{G}_{\mathcal{U}}$ on $\mathcal{U}$ is defined formally as follows.

**Definition 4.2** (Group action on weight spaces)**.** With the notation as above, the *group action* of $\mathcal{G}_{\mathcal{U}}$ on $\mathcal{U}$ is defined to be the map $\mathcal{G}_{\mathcal{U}} \times \mathcal{U} \to \mathcal{U}$ with $(g, U) \mapsto gU = (gW, gb)$, where:

$$(gW)^{(i)} := \left( g^{(i)} \right) \cdot W^{(i)} \cdot \left( g^{(i-1)} \right)^{-1} \quad \text{and} \quad (gb)^{(i)} := \left( g^{(i)} \right) \cdot b^{(i)}. \tag{15}$$

In concrete:

$$(gW)_{jk}^{(i)} := \frac{d_j^{(i)}}{d_k^{(i-1)}} \cdot W_{\pi_i^{-1}(j)\pi_{i-1}^{-1}(k)}^{(i)} \quad \text{and} \quad (gb)_j^{(i)} := d_j^{(i)} \cdot b_{\pi_i^{-1}(j)}^{(i)}. \tag{16}$$

**Remark 4.3.** *The group $\mathcal{G}_{\mathcal{U}}$ is determined only by the number of layers $L$ and the numbers of channels $n_i$, not by the dimensions of weights $w_i$ and biases $b_i$ at each channel.*

The group $\mathcal{G}_{\mathcal{U}}$ has nice behaviors when acting on the weight spaces of FCNNs given in Eq. (5) and CNNs given in Eq. (8). In particular, depending on the specific choice of the activation $\sigma$, the function $f$ built by the given FCNN or CNN is invariant under the action of a subgroup $G$ of $\mathcal{G}_{\mathcal{U}}$, as we will see in the following proposition.

**Proposition 4.4** ($G$-equivariance of neural functionals)**.** *Let $f = f(\,\cdot\,; U, \sigma)$ be an FCNN given in Eq. (5) or CNN given in Eq. (8) with the weight space $U \in \mathcal{U}$ and an activation $\sigma \in \{\mathrm{ReLU}, \mathrm{Tanh}, \sin\}$. Let us defined a subgroup $G$ of $\mathcal{G}_{\mathcal{U}}$ as follows:*

*(i) If $\sigma = \mathrm{ReLU}$, we set $G = \{\mathrm{id}_{\mathcal{G}_{n_L}}\} \times \mathcal{G}_{n_{L-1}}^{>0} \times \ldots \times \mathcal{G}_{n_1}^{>0} \times \{\mathrm{id}_{\mathcal{G}_{n_0}}\}$.*

*(ii) If $\sigma = \sin$ or $\tanh$, then we set $G = \{\mathrm{id}_{\mathcal{G}_{n_L}}\} \times \mathcal{G}_{n_{L-1}}^{\pm 1} \times \ldots \times \mathcal{G}_{n_1}^{\pm 1} \times \{\mathrm{id}_{\mathcal{G}_{n_0}}\}$.*

*Then $f$ is $G$-invariant under the action of $G$ on its weight space, i.e.*

$$f(\mathbf{x}\,;\, U, \sigma) = f(\mathbf{x}\,;\, gU, \sigma) \tag{17}$$

*for all $g \in G$, $U \in \mathcal{U}$ and $\mathbf{x} \in \mathbb{R}^{n_0}$.*

**Remark 4.5** (Maximality of $G$)**.** *The proof of Proposition 4.4 can be found in Appendix C.2. The group $G$ defined above is even proved to be the maximal choice in the case:*

- *$\sigma = \mathrm{ReLU}$ and $n_L \geqslant \ldots \geqslant n_2 \geqslant n_1 > n_0 = 1$ (see [12, 25]), or*
- *$\sigma = \tanh$ (see [14, 22]).*

*Here, $G$ is maximal in the sense that: if $U'$ is another element in $\mathcal{U}$ with $f(\,\cdot\,; U, \sigma) = f(\,\cdot\,; U', \sigma)$, then there exists an element $g \in G$ such that $U' = gU$. It is natural to ask whether the group $G$ is still maximal in the other case. This question still remains open and we leave it for future exploration.*

According to Proposition 4.4, the symmetries of the weight space of an FCNN or CNN must include not only permutation matrices but also other types of monomial matrices resulting from scaling (for ReLU networks) or sign flipping (for $\sin$ and $\tanh$ networks) the weights. Recent works on NFN design only take into account the permutation symmetries of the weight space. Therefore, it is necessary to design a new class of NFNs that incorporates these missing symmetries. We will introduce such a class in the next section.

## 5 Monomial Matrix Group Equivariant and Invariant NFNs

In this section, we introduce a new family of NFNs, called Monomial-NFNs, by incorporating symmetries arising from monomial matrix groups which have been clarified in Proposition 4.4. The main components of Monomial-NFNs are the monomial matrix group equivariant and invariant linear layers between two weight spaces which will be presented in Subsections 5.1 and 5.2, respectively. We will only consider the case of ReLU activation. Network architectures with other activations will be considered in detail in Appendices A and B.

In the following, $\mathcal{U} = (\mathcal{W}, \mathcal{B})$ is the weight space with $L$ layers, $n_i$ channels at $i^{\text{th}}$ layer, and the dimensions of weight and bias are $w_i$ and $b_i$, respectively (see Eqs. (11) and (12)). Since we consider ReLU network architectures, according to Proposition 4.4, the symmetries of the weight space is given by the subgroup $G = \{\mathrm{id}_{\mathcal{G}_{n_L}}\} \times \mathcal{G}_{n_{L-1}}^{>0} \times \ldots \times \mathcal{G}_{n_1}^{>0} \times \{\mathrm{id}_{\mathcal{G}_{n_0}}\}$ of $\mathcal{G}_{\mathcal{U}}$.

Table 1: Number of parameters in a linear equivariant layer $E \colon \mathcal{U} \to \mathcal{U}'$ with respect to permutation matrix groups in [71], and monomial matrix groups. Here, $c = \max\{w_i, b_j\}$ and $c' = \max\{w_i', b_j'\}$.

| Subgroups of $\mathcal{G}_{\mathcal{U}}$ | | Number of parameters of $E$ |
|---|---|---|
| $\mathcal{P}_{n_L} \times \mathcal{P}_{n_{L-1}} \times \ldots \mathcal{P}_{n_1} \times \mathcal{P}_{n_0}$ | ([71]) | $\mathcal{O}(cc'L^2)$ |
| $\{\mathrm{id}_{\mathcal{G}_{n_L}}\} \times \mathcal{P}_{n_{L-1}} \times \ldots \times \mathcal{P}_{n_1} \times \{\mathrm{id}_{\mathcal{G}_{n_0}}\}$ | ([71]) | $\mathcal{O}(cc'(L + n_0 + n_L)^2)$ |
| $\{\mathrm{id}_{\mathcal{G}_{n_L}}\} \times \mathcal{G}_{n_{L-1}}^{>0} \times \ldots \times \mathcal{G}_{n_1}^{>0} \times \{\mathrm{id}_{\mathcal{G}_{n_0}}\}$ | (Ours) | $\mathcal{O}(cc'(L + n_0 + n_L))$ |
| $\{\mathrm{id}_{\mathcal{G}_{n_L}}\} \times \mathcal{G}_{n_{L-1}}^{\pm 1} \times \ldots \times \mathcal{G}_{n_1}^{\pm 1} \times \{\mathrm{id}_{\mathcal{G}_{n_0}}\}$ | (Ours) | $\mathcal{O}(cc'(L + n_0 + n_L))$ |

## 5.1 Equivariant Layers

We now construct a linear $G$-equivariant layer between weight spaces. These layers form the fundamental building blocks for our Monomimal-NFNs. Let $\mathcal{U}$ and $\mathcal{U}'$ be two weight spaces of the same network architecture described in Eqs. (11) and (12), i.e. they have the same number of layers as well as the same number of channels at each layer. Denote the dimension of weights and biases in each channel at the $i$-th layer of $\mathcal{U}'$ as $w_i'$ and $b_i'$, respectively. Note that, in this case, we have $\mathcal{G}_{\mathcal{U}} = \mathcal{G}_{\mathcal{U}'}$. We construct $G$-equivariant afine maps $E \colon \mathcal{U} \to \mathcal{U}'$ with $\mathbf{x} \mapsto \mathfrak{a}\mathbf{x} + \mathfrak{b}$, where $\mathfrak{a} \in \mathbb{R}^{\dim \mathcal{U}' \times \dim \mathcal{U}}$ and $\mathfrak{b} \in \mathbb{R}^{\dim \mathcal{U}' \times 1}$ are learnable parameters.

To make $E$ to be $G$-equivarient, $\mathfrak{a}$ and $\mathfrak{b}$ have to satisfy a system of constraints (usually called *parameter sharing*), which are induced from the condition $E(gU) = gE(U)$ for all $g \in G$ and $U \in \mathcal{U}$. We show in details what are these constraints and how to derive the concrete formula of $E$ in Appendix A. The formula of $E$ is presented as follows: For $U = (W, b) \in \mathcal{U}$, the image $E(U) = (W', b') \in \mathcal{U}'$ is computed by:

$$
\begin{aligned}
W_{jk}'^{(1)} &= \sum_{q=1}^{n_0} \mathfrak{p}_{1jq}^{1jk} W_{jq}^{(1)} + \mathfrak{q}_{1j}^{1jk} b_j^{(1)}, \quad & b_j'^{(1)} &= \sum_{q=1}^{n_0} \mathfrak{r}_{1jq}^{1j} W_{jq}^{(1)} + \mathfrak{s}_{1j}^{1j} b_j^{(1)}, \\
W_{jk}'^{(i)} &= \mathfrak{p}_{ijk}^{ijk} W_{jk}^{(i)}, \quad & b_j'^{(i)} &= \mathfrak{s}_{ij}^{ij} b_j^{(i)}, \quad 1 < i < L, \\
W_{jk}'^{(L)} &= \sum_{p=1}^{n_L} \mathfrak{p}_{Lpk}^{Ljk} W_{pk}^{(L)}, \quad & b_j'^{(L)} &= \sum_{p=1}^{n_L} \mathfrak{s}_{Lp}^{Lj} b_p^{(L)} + \mathfrak{t}^{Lj}.
\end{aligned}
\tag{18}
$$

Here, $(\mathfrak{p}, \mathfrak{q}, \mathfrak{r}, \mathfrak{s}, \mathfrak{t})$ is the hyperparameter of $E$. We discuss in detail the dimensions and sharing information between these parameters in Appendix A.1. Note that, we also show that all linear $G$-equivariant functional are in this form in Appendix A. To conclude, we have:

**Theorem 5.1.** *With notation as above, the linear functional map $E \colon \mathcal{U} \to \mathcal{U}'$ defined by Eq. (18) is $G$-equivariant. Moreover, every $G$-equivariant linear functional map from $\mathcal{U}$ to $\mathcal{U}'$ are in that form.*

**Number of parameters and comparison to previous works.** The number of parameters in our layer is linear in $L, n_0, n_L$, which is significantly smaller than the number of parameters in layers described in [71], where it is quadratic in $L, n_0, n_L$ (see Table 1). This reduction in parameter count means that our model is suitable for weight spaces of large-scale networks and deep NFNs. Intuitively, the advantage of our layer arises because the group $G$ acting on the weight spaces in our setting is much larger, resulting in a significantly smaller number of orbits in the quotient space $\mathcal{U}/G$. Since the number of orbits is equal to the number of parameters, this leads to a more compact representation. Additionally, the presence of the group $\Delta_*^{>0}$ forces many coefficients of the linear layer $E$ to be zero, further contributing to the efficiency of our model.

## 5.2 Invariant Layers

We will construct an $G$-invariant layer $I \colon \mathcal{U} \to \mathbb{R}^d$ for a fixed integer $d > 0$. In order to do that, we will seek a map $I$ in the form:

$$
I = \mathrm{MLP} \circ I_{\mathcal{P}} \circ I_{\Delta^{>0}}, \tag{19}
$$

where $I_{\Delta^{>0}} \colon \mathcal{U} \to \mathcal{U}$ is an $\Delta_*^{>0}$-invariance and $\mathcal{P}_*$-equivariance map, $I_{\mathcal{P}} \colon \mathcal{U} \to \mathbb{R}^{\dim \mathcal{U}}$ is an $\mathcal{P}_*$-invariant map, and $\mathrm{MLP} \colon \mathbb{R}^{\dim \mathcal{U}} \to \mathbb{R}^d$ is an arbitrary multilayer perceptron to adjust the output dimension. Since $G = \mathcal{G}_*^{>0} = \Delta_*^{>0} \rtimes_{\varphi} \mathcal{P}_*$ (see Remark 3.5), the composition $I = \mathrm{MLP} \circ I_{\mathcal{P}} \circ I_{\Delta^{>0}}$ is clearly $G$-invariant as expected. The construction of $I_{\Delta^{>0}}$ and $I_{\mathcal{P}}$ will be presented below.

Table 2: CNN prediction on Tanh subset of Small CNN Zoo with original and augmented data.

| | STATNN | NP | HNP | Monomial-NFN (ours) | Gap |
|---|---|---|---|---|---|
| Original | $0.913 \pm 0.001$ | $0.925 \pm 0.001$ | $\underline{0.933 \pm 0.002}$ | $\mathbf{0.939 \pm 0.001}$ | **0.006** |
| Augmented | $0.914 \pm 0.001$ | $0.928 \pm 0.001$ | $\underline{0.935 \pm 0.001}$ | $\mathbf{0.943 \pm 0.001}$ | **0.008** |

**Construct $I_{\Delta>0}$.** To capture $\Delta_*^{>0}$-invariance, we recall the notion of positively homogeneous of degree zero maps. For $n > 0$, a map $\alpha$ from $\mathbb{R}^n$ is called *positively homogeneous of degree zero* if for all $\lambda > 0$ and $(x_1, \ldots, x_n) \in \mathbb{R}^n$, we have $\alpha(\lambda x_1, \ldots, \lambda x_n) = \alpha(x_1, \ldots, x_n)$. We construct $I_{\Delta>0} : \mathcal{U} \to \mathcal{U}$ by taking collections of positively homogeneous of degree zero functions $\{\alpha_{jk}^{(i)} : \mathbb{R}^{w_i} \to \mathbb{R}^{w_i}\}$ and $\{\alpha_j^{(i)} : \mathbb{R}^{b_i} \to \mathbb{R}^{b_i}\}$, each one corresponds to weight and bias of $\mathcal{U}$. The maps $I_{\Delta>0} : \mathcal{U} \to \mathcal{U}$ that $(W, b) \mapsto (W', b')$ is defined by simply applying these functions on each weight and bias entries as follows:

$$W_{jk}'^{(i)} = \alpha_{jk}^{(i)}(W_{jk}^{(i)}) \text{ and } b_j'^{(i)} = \alpha_j^{(i)}(b_j^{(i)}). \tag{20}$$

$I_{\Delta>0}$ is $\Delta_*^{>0}$-invariant by homogeneity of the $\alpha$ functions. To make it become $\mathcal{P}_*$-equivariant, some $\alpha$ functions have to be shared arross any axis that have permutation symmetry. We derive this relation in Appendix B. Some candidates for positively homogeneous of degree zero functions are also presented in Appendix B. They can be fixed or learnable.

**Construct $I_{\mathcal{P}}$.** To capture $\mathcal{P}_*$-invariance, we simply take summing or averaging the weight and bias across any axis that have permutation symmetry as in [71]. In concrete, we have $I_{\mathcal{P}} : \mathcal{U} \to \mathbb{R}^{\dim \mathcal{U}}$ is computed as follows:

$$I_{\mathcal{P}}(U) = \left( W_{\star,:}^{(1)}, W_{:,\star}^{(L)}, W_{\star,\star}^{(2)}, \ldots, W_{\star,\star}^{(L-1)}; v^{(L)}, v_\star^{(1)}, \ldots, v_\star^{(L-1)} \right). \tag{21}$$

Here, $\star$ denotes summation or averaging over the rows or columns of the weight and bias.

**Remark 5.2.** *In our experiments, we use averaging operator since it is empirically more stable.*

Finally we compose an MLP before $I_{\mathcal{P}}$ and $I_{\Delta>0}$ to obtain an $G$-invariant map. We summarize the above construction as follows.

**Theorem 5.3.** *The functional map $I : \mathcal{U} \to \mathbb{R}^d$ defined by Eq. (19) is $G$-invariant.*

### 5.3 Monomial Matrix Group Equivariant Neural Functionals (Monomial-NFNs)

We build Monomial-NFNs by the constructed equivariant and invariant functional layers, with activations and additional layers discussed below. The equivariant NFN is built by simply stacking $G$-equivariant layers. For the invariant counterpart, we follow the construction in [71]. In particular, we first stack some $G$-equivariant layers, then a $\Delta_*^{>0}$-invariant and $\mathcal{P}_*$-equivariant layer. This makes our NFN to be $\Delta_*^{>0}$-invariant and $\mathcal{P}_*$-equivariant. Then we finish the construction by stacking a $\mathcal{P}_*$-invariant layer and the end. This process makes the whole NFN to be $G$-invariant as expected.

**Activations of $G$-equivariant functionals.** Dealing with equivariance under action of $\mathcal{P}_*$ only requires activation of the NFN is enough, since $\mathcal{P}_*$ acts on only the order of channels in each channel of the weight space. For our $G$-equivariant NFNs, between each layer that is $\Delta_*^{>0}$-equivariant, we have to use the same type of activations as the activation in the network input (i.e. either $\mathrm{ReLU}$, $\sin$ or $\tanh$ in our consideration) to maintain the equivariance of the NFN.

**Fourier Features and Positional Embedding.** As mentioned in [35, 71, 72], Fourier Features [30, 62] and (sinusoidal) position embedding play a significant role in the performance of their functionals. Also, in [71], position embedding breaks the symmetry at input and output neurons, and allows us to use equivariant layers that act on input and output neurons. In our $G$-equivariant layers, we do not consider action on input and output neurons as mentioned. Also, using Fourier Features does not maintain $\Delta_*^{>0}$, so we can not use this Fourier layer for our equivariant Monomial-NFNs, and in our invariant Monomial-NFNs, we only can use Fourier layer after the $\Delta_*^{>0}$-invariant layer. This can be considered as a limitation of Monomial-NFNs.

## 6 Experimental Results

In this session, we empirically demonstrate the performance of our Monomial Matrix Group Equivariant Neural Functional Networks (Monomial-NFNs) on various tasks that are either invariant

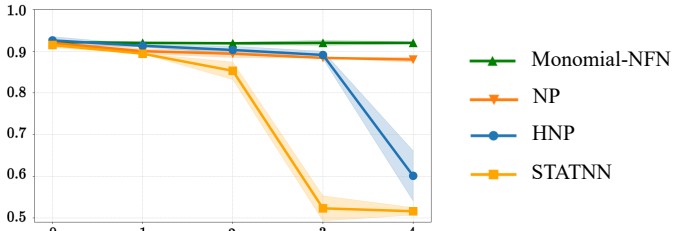

Figure 1: CNN prediction on ReLU subset of Small CNN Zoo with different ranges of augmentations. Here the x-axis is the augment upper scale, presented in log scale. The metric used is Kendall's $\tau$.

Table 3: Classification train and test accuracies (%) for implicit neural representations of MNIST, FashionMNIST, and CIFAR-10. Uncertainties indicate standard error over 5 runs.

|  | Monomial-NFN (ours) | NP | HNP | MLP |
|---|---|---|---|---|
| CIFAR-10 | $\mathbf{34.23 \pm 0.33}$ | $\underline{33.74 \pm 0.26}$ | $31.61 \pm 0.22$ | $10.48 \pm 0.74$ |
| MNIST-10 | $\underline{68.43 \pm 0.51}$ | $\mathbf{69.82 \pm 0.42}$ | $66.02 \pm 0.51$ | $10.62 \pm 0.54$ |
| FashionMNIST | $\mathbf{61.15 \pm 0.55}$ | $\underline{58.21 \pm 0.31}$ | $57.43 \pm 0.46$ | $9.95 \pm 0.36$ |

(predicting CNN generalization from weights and classifying INR representations of images) or equivariant (weight space style editing). We aim to establish two key points. First, our model exhibits more stable behavior when the input undergoes transformations from the monomial matrix groups. Second, our model, Monomial-NFN, achieves competitive performance compared to other baseline models. Our results are averaged over 5 runs. Hyperparameter settings and the number of parameters can be found in Appendix D.

## 6.1 Predicting CNN Generalization from Weights

**Experiment Setup.** In this experiment, we evaluate how our Monomial-NFN predicts the generalization of pretrained CNN networks. We employ the Small CNN Zoo [64], which consists of multiple network weights trained with different initialization and hyperparameter settings, together with activations Tanh or ReLU. Since Monomial-NFNs depend on activations of network inputs, we divide the Small CNN Zoo into two smaller datasets based on their activations. The ReLU dataset considers the group $\mathcal{G}_n^{>0}$, while the Tanh dataset considers the group $\mathcal{G}_n^{\pm 1}$.

We construct the dataset with additional weights that undergo random hidden vector permutation and scaling based on their monomial matrix group. For the ReLU dataset with the group $\mathcal{G}_n^{>0}$, we uniformly sample the diagonal indices of $D$ (see Eq. 14) for various ranges: $[1, 10], [1, 1 \times 10^2], \ldots, [1, 1 \times 10^6]$, while belonging to $\{-1, 1\}$ in the case of Tanh dataset with the group $\mathcal{G}_n^{\pm 1}$. For both datasets, we compare our model with STATNN [65], and with two permutation equivariant neural functional networks from [71], referred to as HNP and NP. To compare the performance of all models, we use Kendall's $\tau$ rank correlation metric [33].

**Results.** We demonstrate the results of all models on the ReLU subset in Figure 1, showing that our model attains stable Kendall's $\tau$ when the scale operators are sampled from different ranges. Specifically, when the log of augmentation upper scale is 0, i.e. the data remains unaltered, our model performs as well as the HNP model. However, as the weights undergo more extensive scaling and permutation, the performance of the HNP and STATNN models drops significantly, indicating their lack of scaling symmetry. The NP model exhibits a similar trend, albeit to a lesser extent. In contrast, our model maintains stable performance throughout.

Table 2 illustrates the performance of all models on both the original and augmented Tanh subsets of CNN Zoo. Our model achieves the highest performance among all models and shows the greatest improvement after training with the augmented dataset. The gap between our model and the second-best model (HNP) increases from 0.006 to 0.008. Additionally, in both experiments, our model utilizes significantly fewer parameters than the baseline models, using only up to $50\%$ of the parameters compared to HNP.

## 6.2 Classifying implicit neural representations of images

**Experiment Setup.** In this experiment, our focus is on extracting the original data information encoded within the weights of implicit neural representations (INRs). We utilize the dataset from [71],

Table 4: Test mean squared error (lower is better) between weight-space editing methods and ground-truth image-space transformations. Uncertainties indicate standard error over 5 runs.

| | Monomial-NFN (ours) | NP | HNP | MLP |
|---|---|---|---|---|
| Contrast (CIFAR-10) | $\mathbf{0.020 \pm 0.001}$ | $\mathbf{0.020 \pm 0.002}$ | $0.021 \pm 0.002$ | $0.031 \pm 0.001$ |
| Dilate (MNIST) | $\underline{0.069 \pm 0.002}$ | $\mathbf{0.068 \pm 0.002}$ | $0.071 \pm 0.001$ | $0.306 \pm 0.001$ |

which comprises pretrained INR networks [58] that encode images from the CIFAR-10 [36], MNIST [39], and FashionMNIST [69] datasets. Each pretrained INR network is designed to map image coordinates $(x, y)$ to color pixel values - 3-dimensional RGB values for CIFAR-10 and 1-dimensional grayscale values for MNIST and FashionMNIST.

**Results.** We compare our model with NP, HNP, and MLP baselines. The results in Table 3 demonstrate that our model outperforms the second-best baseline, NP, for the FashionMNIST and CIFAR-10 datasets by $2.94\%$ and $0.49\%$, respectively. For the MNIST dataset, our model also obtains comparable performance.

### 6.3 Weight space style editing.

**Experiment setup.** In this experiment, we explore altering the weights of the pretrained SIREN model [58] to change the information encoded within the network. We use the network weights provided in the HNP paper for the pretrained SIREN networks on MNIST and CIFAR-10 images. Our focus is on two tasks: the first involves modifying the network to dilate digits from the MNIST dataset, and the second involves altering the SIREN network weights to enhance the contrast of CIFAR-10 images. The objective is to minimize the mean squared error (MSE) training loss between the generated image from the edited SIREN network and the dilated/enhanced contrast image.

**Results.** Table 4 shows that our model performs on par with the best-performing model for increasing the contrast of CIFAR-10 images. For the MNIST digit dilation task, our model also achieves competitive performance compared to the NP baseline. Additionally, Figure 2 presents random samples of the digits that each model encodes for the dilation and contrast tasks, demonstrating that our model's results are visually comparable to those of HNP and NP in both tasks.

## 7 Conclusion

In this paper, we formally describe a group of monomial matrices that preserves FCNNs and CNNs while acting on their weight spaces. For $\mathrm{ReLU}$ networks, this group includes permutation and scaling symmetries, while for networks with $\sin$ or $\mathrm{Tanh}$ activations, it encompasses permutation and sign-flipping symmetries. We introduce Monomial-NFNs, a first-of-a-kind class of NFNs that incorporates these scaling or sign-flipping symmetries in weight spaces. We demonstrate that the low number of trainable parameters in our equivariant linear layer of Monomial-NFNs compared to previous works on NFNs, highlighting their capability to efficiently process weight spaces of deep networks. Our NFNs exhibit competitive generalization performance and efficiency compared to existing models across several benchmarks.

One limitation of our model is that, due to the large size of the group considered, the resulting linear layers can be limited in terms of expressivity. For example, a weight corresponding to an edge between two neurons will be updated based only on its previous value, ignoring other edges across the same or other layers. To resolve this issue, it is necessary to construct an equivariant nonlinear layer to encode further relations between these weights, thus enhancing the expressivity. Another limitation is that we are uncertain about the maximality of the group $G$ acting on the weight space of the ReLU network. Therefore, other types of symmetries may exist in the weight space beyond neuron permutation and weight scaling, and our model is not equivariant with respect to these symmetries. We leave the problem of identifying such a maximal group for future research.

## Acknowledgments and Disclosure of Funding

This research / project is supported by the National Research Foundation Singapore under the AI Singapore Programme (AISG Award No: AISG2-TC-2023-012-SGIL). This research / project is supported by the Ministry of Education, Singapore, under the Academic Research Fund Tier 1 (FY2023) (A-8002040-00-00, A-8002039-00-00). This research / project is also supported by the NUS Presidential Young Professorship Award (A-0009807-01-00).

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

# Supplement to "Monomial Matrix Group Equivariant Neural Functional Networks"

**Table of Contents**

## A   Construction of Monomial Matrix Group Equivariant Layers

In this appendix, we present how we constructed Monomial Matrix Group Equivariant Layers. We adopt the idea of notation in [71] to derive the formula of linear functional layers. For two weight spaces $\mathcal{U}$ and $\mathcal{U}'$ with the same number of layers $L$ as well as the same number of channels at $i$-th layer $n_i$:

$$\mathcal{U} = \mathcal{W} \times \mathcal{B} \quad \text{where:} \tag{22}$$
$$\mathcal{W} = \mathbb{R}^{w_L \times n_L \times n_{L-1}} \times \ldots \times \mathbb{R}^{w_2 \times n_2 \times n_1} \times \mathbb{R}^{w_1 \times n_1 \times n_0},$$
$$\mathcal{B} = \mathbb{R}^{b_L \times n_L \times 1} \times \ldots \times \mathbb{R}^{b_2 \times n_2 \times 1} \times \mathbb{R}^{b_1 \times n_1 \times 1};$$

and

$$\mathcal{U}' = \mathcal{W}' \times \mathcal{B}' \quad \text{where:} \tag{23}$$
$$\mathcal{W}' = \mathbb{R}^{w'_L \times n_L \times n_{L-1}} \times \ldots \times \mathbb{R}^{w'_2 \times n_2 \times n_1} \times \mathbb{R}^{w'_1 \times n_1 \times n_0},$$
$$\mathcal{B}' = \mathbb{R}^{b'_L \times n_L \times 1} \times \ldots \times \mathbb{R}^{b'_2 \times n_2 \times 1} \times \mathbb{R}^{b'_1 \times n_1 \times 1};$$

our equivariant layer $E \colon \mathcal{U} \to \mathcal{U}'$ will has the form as follows:

$$E \quad : (W, b) = U \longmapsto U' = (W', b') \quad \text{where:} \tag{24}$$

$$W_{jk}^{\prime(i)} := \sum_{s=1}^{L} \sum_{p=1}^{n_s} \sum_{q=1}^{n_{s-1}} \mathfrak{p}_{spq}^{ijk} W_{pq}^{(s)} + \sum_{s=1}^{L} \sum_{p=1}^{n_s} \mathfrak{q}_{sp}^{ijk} b_p^{(s)} + \mathfrak{t}^{ijk} \tag{25}$$

$$b_j^{\prime(i)} := \sum_{s=1}^{L} \sum_{p=1}^{n_s} \sum_{q=1}^{n_{s-1}} \mathfrak{r}_{spq}^{ij} W_{pq}^{(s)} + \sum_{s=1}^{L} \sum_{p=1}^{n_s} \mathfrak{s}_{sp}^{ij} b_p^{(s)} + \mathfrak{t}^{ij} \tag{26}$$

Here, the map $E$ is parameterized by hyperparameter $\theta = (\mathfrak{p}, \mathfrak{q}, \mathfrak{s}, \mathfrak{r}, \mathfrak{t})$ with dimensions of each component as follows:

- $\mathfrak{p}_{spq}^{ijk} \in \mathbb{R}^{w'_i \times w_s}$ represents the contribution of $W_{pq}^{(s)}$ to $W_{jk}^{\prime(i)}$,

- $\mathfrak{q}_{sp}^{ijk} \in \mathbb{R}^{w_i' \times b_s}$ represents the contribution of $b_p^{(s)}$ to $W_{jk}'^{(i)}$,

- $\mathfrak{t}^{ijk} \in \mathbb{R}^{w_i'}$ is the bias of the layer for $W_{jk}'^{(i)}$;

- $\mathfrak{r}_{spq}^{ij} \in \mathbb{R}^{b_i' \times w_s}$ represents the contribution of $W_{pq}^{(s)}$ to $b_j'^{(i)}$,

- $\mathfrak{s}_{sp}^{ij} \in \mathbb{R}^{b_i' \times b_s}$ represents the contribution of $b_p^{(s)}$ to $b_j'^{(i)}$,

- $\mathfrak{t}^{ij} \in \mathbb{R}^{b_i'}$ is the bias of the layer for $b_j'^{(i)}$.

We want to see how an element of the group $\mathcal{G}_{\mathcal{U}}$ acts on input and output of layer $E$. Let

$$g = \left(g^{(L)}, \ldots, g^{(0)}\right) \in \mathcal{G}_{n_L} \times \ldots \times \mathcal{G}_{n_0} = \mathcal{G}_{\mathcal{U}}, \tag{27}$$

where

$$g^{(i)} = D^{(i)} \cdot P_{\pi_i} = \operatorname{diag}\left(d_1^{(i)}, \ldots, d_{n_i}^{(i)}\right) \cdot P_{\pi_i} \in \mathcal{G}_{n_i}. \tag{28}$$

Recall the definition of the group action $gU = (gW, gb)$ where:

$$(gW)^{(i)} := \left(g^{(i)}\right) \cdot W^{(i)} \cdot \left(g^{(i-1)}\right)^{-1} \quad \text{and} \quad (gb)^{(i)} := \left(g^{(i)}\right) \cdot b^{(i)}, \tag{29}$$

or in term of entries:

$$(gW)_{jk}^{(i)} := \frac{d_j^{(i)}}{d_k^{(i-1)}} \cdot W_{\pi_i^{-1}(j)\pi_{i-1}^{-1}(k)}^{(i)} \quad \text{and} \quad (gb)_j^{(i)} := d_j^{(i)} \cdot b_{\pi_i^{-1}(j)}^{(i)}. \tag{30}$$

$gE(U) = gU' = (gW', gb')$ is computed as follows:

$$(gW')_{jk}^{(i)} = \frac{d_j^{(i)}}{d_k^{(i-1)}} \cdot W_{\pi_i^{-1}(j)\pi_{i-1}^{-1}(k)}'^{(i)} \tag{31}$$

$$= \frac{d_j^{(i)}}{d_k^{(i-1)}} \cdot \left(\sum_{s=1}^{L}\sum_{p=1}^{n_s}\sum_{q=1}^{n_{s-1}} \mathfrak{p}_{spq}^{i\pi_i^{-1}(j)\pi_{i-1}^{-1}(k)} W_{pq}^{(s)} + \right. \tag{32}$$

$$\left. \sum_{s=1}^{L}\sum_{p=1}^{n_s} \mathfrak{q}_{sp}^{i\pi_i^{-1}(j)\pi_{i-1}^{-1}(k)} b_p^{(s)} + \mathfrak{t}^{i\pi_i^{-1}(j)\pi_{i-1}^{-1}(k)}\right) \tag{33}$$

$$(gb')_j^{(i)} = d_j^{(i)} \cdot b_{\pi_i^{-1}(j)}'^{(i)} \tag{34}$$

$$= d_j^{(i)} \cdot \left(\sum_{s=1}^{L}\sum_{p=1}^{n_s}\sum_{q=1}^{n_{s-1}} \mathfrak{s}_{spq}^{i\pi_i^{-1}(j)} W_{pq}^{(s)} + \right. \tag{35}$$

$$\left. \sum_{s=1}^{L}\sum_{p=1}^{n_s} \mathfrak{r}_{sp}^{i\pi_i^{-1}(j)} b_p^{(s)} + \mathfrak{t}^{i\pi_i^{-1}(j)}\right). \tag{36}$$

$E(gU) = (gU)' = ((gW)', (gU)')$ is computed as follows:

$$(gU)_{jk}^{\prime(i)} = \sum_{s=1}^{L}\sum_{p=1}^{n_s}\sum_{q=1}^{n_{s-1}} \mathfrak{p}_{spq}^{ijk} \cdot \frac{d_p^{(s)}}{d_q^{(s-1)}} \cdot W_{\pi_s^{-1}(p)\pi_{s-1}^{-1}(q)}^{(s)} + \sum_{s=1}^{L}\sum_{p=1}^{n_s} \mathfrak{q}_{sp}^{ijk} \cdot d_p^{(s)} \cdot b_{\pi_s^{-1}(p)}^{(s)} + \mathfrak{t}^{ijk} \quad (37)$$

$$= \sum_{s=1}^{L}\sum_{p=1}^{n_s}\sum_{q=1}^{n_{s-1}} \mathfrak{p}_{s\pi_s(p)\pi_{s-1}(q)}^{ijk} \cdot \frac{d_{\pi_s(p)}^{(s)}}{d_{\pi_{s-1}(q)}^{(s-1)}} \cdot W_{pq}^{(s)} + \sum_{s=1}^{L}\sum_{p=1}^{n_s} \mathfrak{q}_{s\pi_s(p)}^{ijk} \cdot d_{\pi_s(p)}^{(s)} \cdot b_p^{(s)} + \mathfrak{t}^{ijk}$$

$$(38)$$

$$(gb)_j^{\prime(i)} = \sum_{s=1}^{L}\sum_{p=1}^{n_s}\sum_{q=1}^{n_{s-1}} \mathfrak{r}_{spq}^{ij} \cdot \frac{d_p^{(s)}}{d_q^{(s-1)}} \cdot W_{\pi_s^{-1}(p)\pi_{s-1}^{-1}(q)}^{(s)} + \sum_{s=1}^{L}\sum_{p=1}^{n_s} \mathfrak{s}_{sp}^{ij} \cdot d_p^{(s)} \cdot b_{\pi_s^{-1}(p)}^{(s)} + \mathfrak{t}^{ij} \quad (39)$$

$$= \sum_{s=1}^{L}\sum_{p=1}^{n_s}\sum_{q=1}^{n_{s-1}} \mathfrak{r}_{s\pi_s(p)\pi_{s-1}(q)}^{ij} \cdot \frac{d_{\pi_s(p)}^{(s)}}{d_{\pi_{s-1}(q)}^{(s-1)}} \cdot W_{pq}^{(s)} + \sum_{s=1}^{L}\sum_{p=1}^{n_s} \mathfrak{s}_{s\pi_s(p)}^{ij} \cdot d_{\pi_s(p)}^{(s)} \cdot b_p^{(s)} + \mathfrak{t}^{ij}.$$

$$(40)$$

We need $E$ is $G$-equivariant under the action of subgroups of $\mathcal{G}_{\mathcal{U}}$ as in Theorem 4.4. From the above computation, if $gE(U) = E(gU)$, the hyperparameter $\theta = (\mathfrak{p}, \mathfrak{q}, \mathfrak{r}, \mathfrak{s}, \mathfrak{t})$ have to satisfy the system of constraints as follows:

$$\frac{d_j^{(i)}}{d_k^{(i-1)}} \cdot \mathfrak{p}_{spq}^{i\pi_i^{-1}(j)\pi_{i-1}^{-1}(k)} = \mathfrak{p}_{s\pi_s(p)\pi_{s-1}(q)}^{ijk} \cdot \frac{d_{\pi_s(p)}^{(s)}}{d_{\pi_{s-1}(q)}^{(s-1)}} \quad (41)$$

$$\frac{d_j^{(i)}}{d_k^{(i-1)}} \cdot \mathfrak{q}_{sp}^{i\pi_i^{-1}(j)\pi_{i-1}^{-1}(k)} = \mathfrak{q}_{s\pi_s(p)}^{ijk} \cdot d_{\pi_s(p)}^{(s)} \quad (42)$$

$$d_j^{(i)} \cdot \mathfrak{r}_{spq}^{i\pi_i^{-1}(j)} = \mathfrak{r}_{s\pi_s(p)\pi_{s-1}(q)}^{ij} \cdot \frac{d_{\pi_s(p)}^{(s)}}{d_{\pi_{s-1}(q)}^{(s-1)}} \quad (43)$$

$$d_j^{(i)} \cdot \mathfrak{s}_{sp}^{i\pi_i^{-1}(j)} = \mathfrak{s}_{s\pi_s(p)}^{ij} \cdot d_{\pi_s(p)}^{(s)} \quad (44)$$

$$\frac{d_j^{(i)}}{d_k^{(i-1)}} \cdot \mathfrak{t}^{i\pi_i^{-1}(j)\pi_{i-1}^{-1}(k)} = \mathfrak{t}^{ijk} \quad (45)$$

$$d_j^{(i)} \cdot \mathfrak{t}^{i\pi_i^{-1}(j)} = \mathfrak{t}^{ij}. \quad (46)$$

for all possible tuples $((i,j,k),(s,p,q))$ and all $g \in G$. Since the two subgroups $G$ considered in Theorem 4.4 satisfy that: $G \cap \mathcal{P}_i$ is trivial (for $i = 0$ or $i = L$) or the whole $\mathcal{P}_i$ (for $0 < i < L$), so we can simplify the above system of constraints by moving all the permutation $\pi$'s to LHS, then replacing $\pi^{-1}$ by $\pi$. The system, denoted as (*), now is written as follows:

$$\frac{d_j^{(i)}}{d_k^{(i-1)}} \cdot \mathfrak{p}_{s\pi_s(p)\pi_{s-1}(q)}^{i\pi_i(j)\pi_{i-1}(k)} = \mathfrak{p}_{spq}^{ijk} \cdot \frac{d_p^{(s)}}{d_q^{(s-1)}} \quad (*1)$$

$$\frac{d_j^{(i)}}{d_k^{(i-1)}} \cdot \mathfrak{q}_{s\pi_s(p)}^{i\pi_i(j)\pi_{i-1}(k)} = \mathfrak{q}_{sp}^{ijk} \cdot d_p^{(s)} \quad (*2)$$

$$d_j^{(i)} \cdot \mathfrak{r}_{s\pi_s(p)\pi_{s-1}(q)}^{i\pi_i(j)} = \mathfrak{r}_{spq}^{ij} \cdot \frac{d_p^{(s)}}{d_q^{(s-1)}} \quad (*3)$$

$$d_j^{(i)} \cdot \mathfrak{s}_{s\pi_s(p)}^{i\pi_i(j)} = \mathfrak{s}_{sp}^{ij} \cdot d_p^{(s)} \quad (*4)$$

$$\frac{d_j^{(i)}}{d_k^{(i-1)}} \cdot \mathfrak{t}^{i\pi_i^{-1}(j)\pi_{i-1}^{-1}(k)} = \mathfrak{t}^{ijk} \quad (*5)$$

$$d_j^{(i)} \cdot \mathfrak{t}^{i\pi_i^{-1}(j)} = \mathfrak{t}^{ij} \quad (*6)$$

We treat each case of activation separately.

Table 5: Hyperparameter of Equivariant Layers with ReLU activation. *Left* presents all possible case of tuple $((i, j, k), (s, p, q))$, and *Right* presents the parameter at the corresponding position. Here, we have three types of notations: 0 means the parameter equal to 0; equations with $\pi$'s in LHS means the equation holds for all possible $\pi$; and a single term with no further information means the term can be arbitrary.

| Tuple $((i,j,k),(s,p,q))$ | | | Hyperparameter $(\mathfrak{p}, \mathfrak{q}, \mathfrak{r}, \mathfrak{s})$ | | | |
|---|---|---|---|---|---|---|
| $i$ and $s$ | $j$ and $p$ | $k$ and $q$ | $\mathfrak{p}^{ijk}_{spq}$ | $\mathfrak{q}^{ijk}_{sp}$ | $\mathfrak{r}^{ij}_{spq}$ | $\mathfrak{s}^{ij}_{sp}$ |
| $i = s = 1$ | $j \neq p$ | | $0$ | $0$ | $0$ | $0$ |
| | $j = p$ | | $\mathfrak{p}^{1\pi(j)k}_{1\pi(j)q} = \mathfrak{p}^{1jk}_{1jq}$ | $\mathfrak{q}^{1\pi(j)k}_{1\pi(j)} = \mathfrak{q}^{1jk}_{1j}$ | $\mathfrak{r}^{1\pi(j)}_{1\pi(j)q} = \mathfrak{r}^{1j}_{1jq}$ | $\mathfrak{s}^{1\pi(j)}_{1\pi(j)} = \mathfrak{s}^{1j}_{1j}$ |
| $i = s = L$ | | $k \neq q$ | $0$ | $0$ | $0$ | $\mathfrak{s}^{Lj}_{Lp}$ |
| | | $k = q$ | $\mathfrak{p}^{Lj\pi(k)}_{Lp\pi(k)} = \mathfrak{p}^{Ljk}_{Ljq}$ | $0$ | $0$ | $\mathfrak{s}^{Lj}_{Lp}$ |
| $1 < i = s < L$ | $j \neq p$ | | $0$ | $0$ | $0$ | $0$ |
| | $j = p$ | $k \neq q$ | $0$ | $0$ | $0$ | $\mathfrak{s}^{i\pi(j)}_{i\pi(j)} = \mathfrak{s}^{ij}_{ij}$ |
| | | $k = q$ | $\mathfrak{p}^{i\pi(j)\pi'(k)}_{i\pi(j)\pi'(k)} = \mathfrak{p}^{ijk}_{ijk}$ | $0$ | $0$ | $\mathfrak{s}^{i\pi(j)}_{i\pi(j)} = \mathfrak{s}^{ij}_{ij}$ |
| $i \neq s$ | | | $0$ | $0$ | $0$ | $0$ |

Table 6: Construction of equivariant functional layer with ReLU activation. Note that all parameters have to satisfy the conditions presented in Table 5.

| Layer | Equivariant layer $E : (W, b) \longmapsto (W', b')$ | |
|---|---|---|
| | $W'^{(i)}_{jk}$ | $b'^{(i)}_j$ |
| $i = 1$ | $\sum_{q=1}^{n_0} \mathfrak{p}^{1jk}_{1jq} W^{(1)}_{jq} + \mathfrak{q}^{1jk}_{1j} b^{(1)}_j$ | $\sum_{q=1}^{n_0} \mathfrak{r}^{1j}_{1jq} W^{(1)}_{jq} + \mathfrak{s}^{1j}_{1j} b^{(1)}_j$ |
| $1 < i < L$ | $\mathfrak{p}^{ijk}_{ijk} W^{(i)}_{jk}$ | $\mathfrak{s}^{ij}_{ij} b^{(i)}_j$ |
| $i = L$ | $\sum_{p=1}^{n_L} \mathfrak{p}^{Ljk}_{Lpk} W^{(L)}_{pk}$ | $\sum_{p=1}^{n_L} \mathfrak{s}^{Lj}_{Lp} b^{(L)}_p + \mathfrak{t}^{Lj}$ |

## A.1 ReLU activation

Recall that, in this case:

$$G := \{\mathrm{id}_{\mathcal{G}_{n_L}}\} \times \mathcal{G}^{>0}_{n_{L-1}} \times \ldots \times \mathcal{G}^{>0}_{n_1} \times \{\mathrm{id}_{\mathcal{G}_{n_0}}\}. \tag{47}$$

So the system of constraints (*) holds for:

1. all possible tuples $((i, j, k), (s, p, q))$,

2. all $\pi_i \in \mathcal{P}_i$ for $0 < i < L$, all $d^{(i)}_j > 0$ for $0 < i < L, 1 \leqslant j \leqslant n_i$,

3. $\pi_i = \mathrm{id}_{\mathcal{G}_{n_i}}$ and $d^{(i)}_j = 1$ for $i = 0$ or $i = L$.

By treat each case of tuples $((i, j, k), (s, p, q))$, we solve Eq. *1, Eq. *2, Eq. *3, Eq. *4 in the system (*) for hyperparameter $(\mathfrak{p}, \mathfrak{q}, \mathfrak{r}, \mathfrak{s})$ as in Table 5. For $\mathfrak{t}^{ijk}$ and $\mathfrak{t}^{ij}$, by Eq. *5, Eq. *6, we have $\mathfrak{t}^{ijk} = 0$ for all $(i, j, k)$, $\mathfrak{t}^{ij} = 0$ if $i < L$, and $\mathfrak{t}^{Lj}$ is arbitrary for all $1 \leqslant j \leqslant n_L$. In conclusion, the formula of equivariant layers $E$ in case of activation ReLU is presented as in Table 6.

**Example A.1.** Let us consider a two-hidden-layers MLP with activation $\sigma = \mathrm{ReLU}$. Assume that $n_0 = n_1 = n_2 = n_3 = 2$, i.e., all layers have two neurons. This MLP defines a function $f : \mathbb{R}^2 \to \mathbb{R}^2$ given by

$$f(x) = W^{(3)} \sigma \left( W^{(2)} \sigma \left( W^{(1)} x + b^{(1)} \right) + b^{(2)} \right) + b^{(3)},$$

where $W^{(i)} = \begin{pmatrix} W_{11}^{(i)} & W_{12}^{(i)} \\ W_{21}^{(i)} & W_{22}^{(i)} \end{pmatrix}$ is a $2 \times 2$ matrix and $b^{(i)} = \begin{bmatrix} b_1^{(i)} \\ b_2^{(i)} \end{bmatrix}$ for each $i = 1, 2, 3$. In this case, the weight space $\mathcal{U}$ consists of the tuples

$$U = (W^{(1)}, W^{(2)}, W^{(3)}, b^{(1)}, b^{(2)}, b^{(3)})$$

and it has dimension 18.

According to Eq. (27), an equivariant layer $E$ over $\mathcal{U}$ has the form

$$E(U) = \left( W'^{(1)}, W'^{(2)}, W'^{(3)}, b'^{(1)}, b'^{(2)}, b'^{(3)} \right),$$

where

$$W_{jk}'^{(1)} = \mathfrak{p}_{1j_1}^{1jk} W_{j_1 1}^{(1)} + \mathfrak{p}_{1j_2}^{1jk} W_{j_2 2}^{(1)} + \mathfrak{q}_{1j}^{1jk} b_j^{(1)}, \qquad b_j'^{(1)} = \mathfrak{r}_{j_1}^{1j} W_{j_1 1}^{(1)} + \mathfrak{r}_{j_2}^{1j} W_{j_2 2}^{(1)} + \mathfrak{s}_{1j}^{1j} b_j^{(1)},$$

$$W_{jk}'^{(2)} = \mathfrak{p}_{2j}^{2jk} W_{jk}^{(2)}, \qquad b_j'^{(2)} = \mathfrak{s}_{2j}^{2j} b_j^{(2)},$$

$$W_{jk}'^{(3)} = \mathfrak{p}_{3k_1}^{3jk} W_{3k}^{(3)} + \mathfrak{p}_{3k_2}^{3jk} W_{2k}^{(3)}, \qquad b_j'^{(3)} = \mathfrak{s}_{3j_1}^{3j} b_1^{(3)} + \mathfrak{s}_{3j_2}^{3j} b_2^{(3)} + \mathfrak{r}_j^3.$$

These equations can be written in a friendly matrix form as follows.

$$\begin{bmatrix} W_{11}'^{(1)} \\ W_{12}'^{(1)} \\ W_{21}'^{(1)} \\ W_{22}'^{(1)} \\ b_1'^{(1)} \\ b_2'^{(1)} \end{bmatrix} = \begin{bmatrix} \mathfrak{p}_{111}^{111} & \mathfrak{p}_{112}^{111} & 0 & 0 & \mathfrak{q}_{111}^{111} & 0 \\ \mathfrak{p}_{111}^{112} & \mathfrak{p}_{112}^{112} & 0 & 0 & \mathfrak{q}_{111}^{112} & 0 \\ 0 & 0 & \mathfrak{p}_{121}^{121} & \mathfrak{p}_{122}^{121} & 0 & \mathfrak{q}_{112}^{121} \\ 0 & 0 & \mathfrak{p}_{121}^{122} & \mathfrak{p}_{122}^{122} & 0 & \mathfrak{q}_{112}^{122} \\ \mathfrak{r}_{111}^{111} & \mathfrak{r}_{112}^{111} & 0 & 0 & \mathfrak{s}_{111}^{111} & 0 \\ 0 & 0 & \mathfrak{r}_{121}^{121} & \mathfrak{r}_{122}^{122} & 0 & \mathfrak{s}_{112}^{112} \end{bmatrix} \begin{bmatrix} W_{11}^{(1)} \\ W_{12}^{(1)} \\ W_{21}^{(1)} \\ W_{22}^{(1)} \\ b_1^{(1)} \\ b_2^{(1)} \end{bmatrix},$$

$$\begin{bmatrix} W_{11}'^{(2)} \\ W_{12}'^{(2)} \\ W_{21}'^{(2)} \\ W_{22}'^{(2)} \\ b_1'^{(2)} \\ b_2'^{(2)} \end{bmatrix} = \begin{bmatrix} \mathfrak{p}_{211}^{211} & 0 & 0 & 0 & 0 & 0 \\ 0 & \mathfrak{p}_{212}^{212} & 0 & 0 & 0 & 0 \\ 0 & 0 & \mathfrak{p}_{221}^{221} & 0 & 0 & 0 \\ 0 & 0 & 0 & \mathfrak{p}_{222}^{222} & 0 & 0 \\ 0 & 0 & 0 & 0 & \mathfrak{s}_{211}^{211} & 0 \\ 0 & 0 & 0 & 0 & 0 & \mathfrak{s}_{222}^{222} \end{bmatrix} \begin{bmatrix} W_{11}^{(2)} \\ W_{12}^{(2)} \\ W_{21}^{(2)} \\ W_{22}^{(2)} \\ b_1^{(2)} \\ b_2^{(2)} \end{bmatrix},$$

$$\begin{bmatrix} W_{11}'^{(3)} \\ W_{12}'^{(3)} \\ W_{21}'^{(3)} \\ W_{22}'^{(3)} \\ b_1'^{(3)} \\ b_2'^{(3)} \end{bmatrix} = \begin{bmatrix} \mathfrak{p}_{311}^{311} & 0 & \mathfrak{p}_{321}^{311} & 0 & 0 & 0 \\ 0 & \mathfrak{p}_{312}^{312} & 0 & \mathfrak{p}_{322}^{322} & 0 & 0 \\ \mathfrak{p}_{312}^{321} & 0 & \mathfrak{p}_{321}^{321} & 0 & 0 & 0 \\ 0 & \mathfrak{p}_{312}^{322} & 0 & \mathfrak{p}_{322}^{322} & 0 & 0 \\ 0 & 0 & 0 & 0 & \mathfrak{s}_{311}^{311} & \mathfrak{s}_{312}^{312} \\ 0 & 0 & 0 & 0 & \mathfrak{s}_{321}^{321} & \mathfrak{s}_{322}^{322} \end{bmatrix} \begin{bmatrix} W_{11}^{(3)} \\ W_{12}^{(3)} \\ W_{21}^{(3)} \\ W_{22}^{(3)} \\ b_1^{(3)} \\ b_2^{(3)} \end{bmatrix} + \begin{bmatrix} 0 \\ 0 \\ 0 \\ 0 \\ \mathfrak{r}_1^3 \\ \mathfrak{r}_2^3 \end{bmatrix}.$$

## A.2 Sin or Tanh activation

Recall that, in this case:

$$G := \{\mathrm{id}_{\mathcal{G}_{n_L}}\} \times \mathcal{G}_{n_{L-1}}^{\pm 1} \times \ldots \times \mathcal{G}_{n_1}^{\pm 1} \times \{\mathrm{id}_{\mathcal{G}_{n_0}}\}. \tag{48}$$

So the system of constraints (*) holds for:

1. all possible tuples $((i, j, k), (s, p, q))$,

2. all $\pi_i \in \mathcal{P}_i$ for $0 < i < L$, all $d_j^{(i)} \in \{\pm 1\}$ for $0 < i < L$, $1 \leqslant j \leqslant n_i$,

3. $\pi_i = \mathrm{id}_{\mathcal{G}_{n_i}}$ and $d_j^{(i)} = 1$ for $i = 0$ or $i = L$.

We assume $L \geqslant 3$, the case $L \leqslant 2$ can be solved similarly. By treat each case of tuples $((i, j, k), (s, p, q))$, we solve Eq. *1, Eq. *2, Eq. *3, Eq. *4 in the system (*) for hyperparameter $(\mathfrak{p}, \mathfrak{q}, \mathfrak{r}, \mathfrak{s})$ as in Table 7. For $\mathfrak{t}^{ijk}$ and $\mathfrak{t}^{ij}$, by Eq. *5, Eq. *6, we have $\mathfrak{t}^{ijk} = 0$ for all $(i, j, k)$, $\mathfrak{t}^{ij} = 0$ if $i < L$, and $\mathfrak{t}^{Lj}$ is arbitrary for all $1 \leqslant j \leqslant n_L$. In conclusion, the formula of equivariant layers $E$ in case of sin or Tanh activation is presented as in Table 8.

Table 7: Hyperparameter of Equivariant Layers with $\sin$ or $\mathrm{Tanh}$ activation. *Left* presents all possible case of tuple $((i,j,k),(s,p,q))$, and *Right* presents the parameter at the corresponding position. Here, we have three types of notations: 0 means the parameter equal to 0; equations with $\pi$'s in LHS means the equation holds for all possible $\pi$; and a single term with no further information means the term can be arbitrary.

| Tuple $((i,j,k),(s,p,q))$ | | | Hyperparameter $(\mathfrak{p},\mathfrak{q},\mathfrak{r},\mathfrak{s})$ | | | |
|---|---|---|---|---|---|---|
| $i$ and $s$ | $j$ and $p$ | $k$ and $q$ | $\mathfrak{p}^{ijk}_{spq}$ | $\mathfrak{q}^{ijk}_{sp}$ | $\mathfrak{r}^{ij}_{spq}$ | $\mathfrak{s}^{ij}_{sp}$ |
| $i=s=1$ | $j \neq p$ | | 0 | 0 | 0 | 0 |
| | $j=p$ | | $\mathfrak{p}^{1\pi(j)k}_{1\pi(j)q}=\mathfrak{p}^{1jk}_{1jq}$ | $\mathfrak{q}^{1\pi(j)k}_{1\pi(j)}=\mathfrak{q}^{1jk}_{1j}$ | $\mathfrak{r}^{1\pi(j)}_{1\pi(j)q}=\mathfrak{r}^{1j}_{1jq}$ | $\mathfrak{s}^{1\pi(j)}_{1\pi(j)}=\mathfrak{s}^{1j}_{1j}$ |
| $i=s=L$ | | $k \neq q$ | 0 | 0 | 0 | $\mathfrak{s}^{Lj}_{Lp}$ |
| | | $k=q$ | $\mathfrak{p}^{Lj\pi(k)}_{Lp\pi(k)}=\mathfrak{p}^{Ljk}_{Ljq}$ | 0 | 0 | $\mathfrak{s}^{Lj}_{Lp}$ |
| $1<i=s<L$ | $j \neq p$ | | 0 | 0 | 0 | 0 |
| | $j=p$ | $k \neq q$ | 0 | 0 | 0 | $\mathfrak{s}^{i\pi(j)}_{i\pi(j)}=\mathfrak{s}^{ij}_{ij}$ |
| | | $k=q$ | $\mathfrak{p}^{i\pi(j)\pi'(k)}_{i\pi(j)\pi'(k)}=\mathfrak{p}^{ijk}_{ijk}$ | 0 | 0 | $\mathfrak{s}^{i\pi(j)}_{i\pi(j)}=\mathfrak{s}^{ij}_{ij}$ |
| $(i,s)=(L-1,L)$ | $j=q$ | | 0 | 0 | $\mathfrak{r}^{(L-1)\pi(j)}_{Lp\pi(j)}=\mathfrak{r}^{(L-1)j}_{Lpj}$ | 0 |
| $(i,s)=(L,L-1)$ | $k=p$ | | 0 | $\mathfrak{q}^{Lj\pi(k)}_{(L-1)\pi(k)}=\mathfrak{q}^{Ljk}_{(L-1)k}$ | 0 | 0 |
| otherwise | | | 0 | 0 | 0 | 0 |

Table 8: Construction of equivariant functional layer with $\sin$ or $\mathrm{Tanh}$ activation. Note that all parameters have to satisfy the conditions presented in Table 5.

| Layer | Equivariant layer $E : (W,b) \longmapsto (W',b')$ | |
|---|---|---|
| | $W'^{(i)}_{jk}$ | $b'^{(i)}_j$ |
| $i=1$ | $\sum_{q=1}^{n_0} \mathfrak{p}^{1jk}_{1jq}W^{(1)}_{jq} + \mathfrak{q}^{1jk}_{1j}b^{(1)}_j$ | $\sum_{q=1}^{n_0} \mathfrak{r}^{1j}_{1jq}W^{(1)}_{jq} + \mathfrak{s}^{1j}_{1j}b^{(1)}_j$ |
| $1<i<L-1$ | $\mathfrak{p}^{ijk}_{ijk}W^{(i)}_{jk}$ | $\mathfrak{s}^{ij}_{ij}b^{(i)}_j$ |
| $i=L-1$ | $\mathfrak{p}^{(L-1)jk}_{(L-1)jk}W^{(L-1)}_{jk}$ | $\sum_{p=1}^{n_L} \mathfrak{r}^{(L-1)j}_{Lpj}W^{(L)}_{pj} + \mathfrak{s}^{(L-1)j}_{(L-1)j}b^{(L-1)}_j$ |
| $i=L$ | $\sum_{p=1}^{n_L} \mathfrak{p}^{Ljk}_{Lpk}W^{(L)}_{pk} + \mathfrak{q}^{Ljk}_{(L-1)k}b^{(L-1)}_k$ | $\sum_{p=1}^{n_L} \mathfrak{s}^{Lj}_{Lp}b^{(L)}_p + \mathfrak{t}^{Lj}$ |

## B  Construction of Monomial Matrix Group Invariant Layers

In this appendix, we present how we constructed Monomial Matrix Group Invariant Layers. Let $\mathcal{U}$ be a weight spaces with the number of layers $L$ as well as the number of channels at $i$-th layer $n_i$. We want to construct $G$-invariant layers $I : \mathcal{U} \to \mathbb{R}^d$ for some $d > 0$. We treat each case of activations separately.

### B.1  ReLU activation

Recall that, in this case:

$$G := \{\mathrm{id}_{\mathcal{G}_{n_L}}\} \times \mathcal{G}^{\pm 1}_{n_{L-1}} \times \ldots \times \mathcal{G}^{\pm 1}_{n_1} \times \{\mathrm{id}_{\mathcal{G}_{n_0}}\}. \tag{49}$$

Since $\mathcal{G}^{>0}_*$ is the semidirect product of $\Delta^{>0}_*$ and $\mathcal{P}_*$ with $\Delta^{>0}_*$ is the normal subgroup, we will treat these two actions consecutively, $\Delta^{>0}_*$ first then $\mathcal{P}_*$. We denote these layers by $I_{\Delta^{>0}}$ and $I_{\mathcal{P}}$. Note that, since $I_{\Delta^{>0}}$ comes before $I_{\mathcal{P}}$, $I_{\Delta^{>0}}$ is required to be $\Delta^{>0}_*$-invariant and $\mathcal{P}_*$-equivariant, and $I_{\mathcal{P}}$ is required to be $\mathcal{P}_*$-invariant.

$\Delta^{>0}_*$-**invariance and** $\mathcal{P}_*$-**equivariance.**   To capture $\Delta^{>0}_*$-invariance, we recall the notion of positively homogeneous of degree zero maps. For $n > 0$, a map $\alpha$ from $\mathbb{R}^n$ is called *positively*

Table 9: Constraints of $\alpha$ component in invariant functional layer with ReLU, sin, Tanh activations.

| Layer | $I_{\Delta>0}: (W, b) \longmapsto (W', b')$ | |
| --- | --- | --- |
| | $\alpha_{jk}^{(i)}: W_{jk}^{(i)}) \longmapsto W_{jk}'^{(i)}$ | $\alpha_j^{(i)}: b_j^{(i)} \longmapsto b_j'^{(i)}$ |
| $i = 1$ | $\alpha_{\pi(j)k}^{(i)} = \alpha_{jk}^{(i)}$ | $\alpha_{\pi(j)}^{(i)} = \alpha_j^{(i)}$ |
| $1 < i < L$ | $\alpha_{\pi(j)\pi'(k)}^{(i)} = \alpha_{jk}^{(i)}$ | $\alpha_{\pi(j)}^{(i)} = \alpha_j^{(i)}$ |
| $i = L$ | $\alpha_{j\pi(k)}^{(i)} = \alpha_{jk}^{(i)}$ | $\alpha_j^{(i)}$ |

*homogeneous of degree zero* if

$$\alpha(\lambda x_1, \ldots, \lambda x_n) = \alpha(x_1, \ldots, x_n). \tag{50}$$

for all $\lambda > 0$ and $(x_1, \ldots, x_n) \in \mathbb{R}^n$. We construct $I_{\Delta>0}: \mathcal{U} \to \mathcal{U}$ by taking collections of positively homogeneous of degree zero functions $\{\alpha_{jk}^{(i)}: \mathbb{R}^{w_i} \to \mathbb{R}^{w_i}\}$ and $\{\alpha_j^{(i)}: \mathbb{R}^{b_i} \to \mathbb{R}^{b_i}\}$, each one corresponds to weight and bias of $\mathcal{U}$. The maps $I_{\Delta>0}: \mathcal{U} \to \mathcal{U}$ that $(W, b) \mapsto (W', b')$ is defined by simply applying these functions on each weight and bias entries as follows:

$$W_{jk}'^{(i)} = \alpha_{jk}^{(i)}(W_{jk}^{(i)}) \quad \text{and} \quad b_j'^{(i)} = \alpha_j^{(i)}(b_j^{(i)}). \tag{51}$$

$I_{\Delta>0}$ is $\Delta_*^{>0}$-invariant by homogeneity of the $\alpha$ functions. To make it become $\mathcal{P}_*$-equivariant, some $\alpha$ functions have to be shared arross any axis that have permutation symmetry, presented in Table 9.

**Candidates of function $\alpha$.** We simply choose positively homogeneous of degree zero function $\alpha: \mathbb{R}^n \to \mathbb{R}^n$ by taking $\alpha(0) = 0$ and:

$$\alpha(x_1, \ldots, x_n) = \beta\left(\frac{x_1^2}{x_1^2 + \ldots + x_n^2}, \ldots, \frac{x_n^2}{x_1^2 + \ldots + x_n^2}\right). \tag{52}$$

where $\beta: \mathbb{R}^n \to \mathbb{R}^n$ is an arbitrary function. The function $\beta$ can be fixed or parameterized to make $\alpha$ to be fixed or learnable.

**$\mathcal{P}_*$-invariance.** To capture $\mathcal{P}_*$-invariance, we simply take summing or averaging the weight and bias across any axis that have permutation symmetry as in [71]. In concrete, some $d > 0$, we have $I_{\mathcal{P}}: \mathcal{U} \to \mathbb{R}^d$ is computed as follows:

$$I_{\mathcal{P}}(U) = \left(W_{\star,:}^{(1)}, W_{:,\star}^{(L)}, W_{\star,\star}^{(2)}, \ldots, W_{\star,\star}^{(L-1)}; v^{(L)}, v_\star^{(1)}, \ldots, v_\star^{(L-1)}\right). \tag{53}$$

Here, $\star$ denotes summation or averaging over the rows or columns of the weight and bias.

**$G$−invariance.** Now we simply compose $I_{\mathcal{P}} \circ I_{\Delta>0}$ to get an $G$-invariant map. We use an MLP to complete constructing an $G$-invariant layer with output dimension $d$ as desired:

$$I = \text{MLP} \circ I_{\mathcal{P}} \circ I_{\Delta>0}. \tag{54}$$

### B.2 Sin or Tanh activation

Recall that, in this case:

$$G := \{\text{id}_{\mathcal{G}_{n_L}}\} \times \mathcal{G}_{n_{L-1}}^{\pm 1} \times \ldots \times \mathcal{G}_{n_1}^{\pm 1} \times \{\text{id}_{\mathcal{G}_{n_0}}\}. \tag{55}$$

Since $\mathcal{G}_*^{\pm 1}$ is the semidirect product of $\Delta_*^{\pm 1}$ and $\mathcal{P}_*$ with $\Delta_*^{\pm 1}$ is the normal subgroup, we will treat these two actions consecutively, $\Delta_*^{\pm 1}$ first then $\mathcal{P}_*$. We denote these layers by $I_{\Delta^{\pm 1}}$ and $I_{\mathcal{P}}$. Note that, since $I_{\Delta^{\pm 1}}$ comes before $I_{\mathcal{P}}$, $I_{\Delta^{\pm 1}}$ is required to be $\Delta_*^{\pm 1}$-invariant and $\mathcal{P}_*$-equivariant, and $I_{\mathcal{P}}$ is required to be $\mathcal{P}_*$-invariant.

$\Delta_*^{\pm 1}$**-invariance and** $\mathcal{P}_*$**-equivariance.** To capture $\Delta_*^{\pm 1}$-invariance, we use even functions, i.e. $\alpha(x) = \alpha(-x)$ for all $x$. We construct $I_{\Delta^{\pm 1}} \colon \mathcal{U} \to \mathcal{U}$ by taking collections of even functions $\{\alpha_{jk}^{(i)} \colon \mathbb{R}^{w_i} \to \mathbb{R}^{w_i}\}$ and $\{\alpha_j^{(i)} \colon \mathbb{R}^{b_i} \to \mathbb{R}^{b_i}\}$, each one corresponds to weight and bias of $\mathcal{U}$. The maps $I_{\Delta^{\pm 1}} \colon \mathcal{U} \to \mathcal{U}$ that $(W, b) \mapsto (W', b')$ is defined by simply applying these functions on each weight and bias entries as follows:

$$W_{jk}'^{(i)} = \alpha_{jk}^{(i)}(W_{jk}^{(i)}) \text{ and } b_j'^{(i)} = \alpha_j^{(i)}(b_j^{(i)}). \tag{56}$$

$I_{\Delta^{\pm 1}}$ is $\Delta_*^{\pm 1}$-invariant by design. To make it become $\mathcal{P}_*$-equivariant, some $\alpha$ functions have to be shared arross any axis that have permutation symmetry, presented in Table 9.

**Candidates of function** $\alpha$**.** We simply choose even function $\alpha \colon \mathbb{R}^n \to \mathbb{R}^n$ by:

$$\alpha(x_1, \ldots, x_n) = \beta(|x_1|, \ldots, |x_n|). \tag{57}$$

where $\beta \colon \mathbb{R}^n \to \mathbb{R}^n$ is an arbitrary function. The function $\beta$ can be fixed or parameterized to make $\alpha$ to be fixed or learnable.

$\mathcal{P}_*$**-invariance.** To capture $\mathcal{P}_*$-invariance, we simply take summing or averaging the weight and bias across any axis that have permutation symmetry as in [71]. In concrete, some $d > 0$, we have $I_{\mathcal{P}} \colon \mathcal{U} \to \mathbb{R}^d$ is computed as follows:

$$I_{\mathcal{P}}(U) = \left( W_{\star, :}^{(1)}, W_{:, \star}^{(L)}, W_{\star, \star}^{(2)}, \ldots, W_{\star, \star}^{(L-1)}; v^{(L)}, v_\star^{(1)}, \ldots, v_\star^{(L-1)} \right). \tag{58}$$

Here, $\star$ denotes summation or averaging over the rows or columns of the weight and bias.

$G-$**invariance.** Now we simply compose $I_{\mathcal{P}} \circ I_{\Delta^{\pm 1}}$ to get an $G$-invariant map. We use an MLP to complete constructing an $G$-invariant layer with output dimension $d$ as desired:

$$I = \text{MLP} \circ I_{\mathcal{P}} \circ I_{\Delta^{\pm 1}}. \tag{59}$$

## C Proofs of Theoretical Results

### C.1 Proof of Proposition 3.4

*Proof.* We simply denote the activation ReLU or sin or tanh by $\sigma$. Let $A \in \text{GL}(n)$ that satisfies:

$$\sigma(A \cdot \mathbf{x}) = A \cdot \sigma(\mathbf{x}),$$

for all $\mathbf{x} \in \mathbb{R}^n$. This means:

$$\sigma\left( \begin{bmatrix} a_{11} & \cdots & a_{1n} \\ \vdots & \ddots & \vdots \\ a_{n1} & \cdots & a_{nn} \end{bmatrix} \cdot \begin{bmatrix} x_1 \\ \vdots \\ x_n \end{bmatrix} \right) = \begin{bmatrix} a_{11} & \cdots & a_{1n} \\ \vdots & \ddots & \vdots \\ a_{n1} & \cdots & a_{nn} \end{bmatrix} \cdot \sigma\left( \begin{bmatrix} x_1 \\ \vdots \\ x_n \end{bmatrix} \right),$$

for all $x_1, \ldots, x_n \in \mathbb{R}$. We rewrite this equation as:

$$\sigma\left( \begin{bmatrix} a_{11}x_1 + a_{12}x_2 + \ldots + a_{1n}x_n \\ \vdots \\ a_{n1}x_1 + a_{n2}x_2 + \ldots + a_{nn}x_n \end{bmatrix} \right) = \begin{bmatrix} a_{11} & \cdots & a_{1n} \\ \vdots & \ddots & \vdots \\ a_{n1} & \cdots & a_{nn} \end{bmatrix} \cdot \begin{bmatrix} \sigma(x_1) \\ \vdots \\ \sigma(x_n) \end{bmatrix},$$

or equivalently:

$$\begin{bmatrix} \sigma(a_{11}x_1 + a_{12}x_2 + \ldots + a_{1n}x_n) \\ \vdots \\ \sigma(a_{n1}x_1 + a_{n2}x_2 + \ldots + a_{nn}x_n) \end{bmatrix} = \begin{bmatrix} a_{11}\sigma(x_1) + a_{12}\sigma(x_2) + \ldots + a_{1n}\sigma(x_n) \\ \vdots \\ a_{n1}\sigma(x_1) + a_{n2}\sigma(x_2) + \ldots + a_{nn}\sigma(x_n) \end{bmatrix}.$$

Thus,

$$\sigma\left( \sum_{j=1}^n a_{ij}x_j \right) = \sum_{j=1}^n a_{ij}\sigma(x_j),$$

for all $x_1, \ldots, x_n \in \mathbb{R}$ and $i = 1, \ldots, n$. We will consider the case $i = 1$, i.e.

$$\sigma\left( \sum_{j=1}^n a_{1j}x_j \right) = \sum_{j=1}^n a_{1j}\sigma(x_j), \tag{60}$$

and treat the case $i > 1$ similarly. Now we consider the activation $\sigma$ case by case as follows.

(i) **Case 1.** $\sigma = \mathrm{ReLU}$. We have some observations:

1. Let $x_1 = 1$, and $x_2 = \ldots = x_n = 0$. Then from Eq. (60), we have:
$$\sigma(a_{11}) = a_{11},$$
which implies that $a_{11} \geqslant 0$. Similarly, we also have $a_{12}, \ldots, a_{1n} \geqslant 0$.

2. Since $A$ is an invertible matrix, the entries $a_{11}, \ldots, a_{1n}$ in the first row of $A$ can not be simultaneously equal to $0$.

3. There is at most only one nonzero number among the entries $a_{11}, \ldots, a_{1n}$. Indeed, assume by the contrary that $a_{11}, a_{12} > 0$. Let $x_3 = \ldots = x_n = 0$, from Eq. (60), we have:
$$\sigma(a_{11}x_1 + a_{12}x_2) = a_{11}\sigma(x_1) + a_{12}\sigma(x_2).$$
Let $x_2 = -1$, we have:
$$\sigma(a_{11}x_1 - a_{12}) = a_{11}\sigma(x_1).$$
Now, let $x_1 > 0$ be a sufficiently large number such that $a_{11}x_1 - a_{12} > 0$. (Note that this number exists since $a_{11} > 0$). Then we have:
$$a_{11}x_1 - a_{12} = a_{11}x_1,$$
which implies $a_{12} = 0$, a contradiction.

It follows from these three observations that there is exactly one non-zero element among the entries $a_{11}, \ldots, a_{1n}$. In other words, matrix $A$ has exactly one nonzero entry in the first row. This applies for every row, so $A$ has exactly one non-zero entry in each row. Since $A$ is invertible, each column of $A$ has at least one non-zero entry. Thus $A$ also has exactly one non-zero entry in each column. Hence, $A$ is in $\mathcal{G}_n$. Moreover, all entries of $A$ are non-negative, so $A$ is in $\mathcal{G}_n^{>0}$.

It is straight forward to check that for all $A$ in $\mathcal{G}_n^{>0}$ we have $\sigma(A \cdot \mathbf{x}) = A \cdot \sigma(\mathbf{x})$.

(ii) **Case 2.** $\sigma = \mathrm{Tanh}$ or $\sigma = \sin$. We have some observations:

1. Let $x_2 = \ldots = x_n = 0$. Then from Eq. (60), we have:
$$\sigma(a_{11}x_1) = a_{11}\sigma(x_1),$$
which implies $a_{11} \in \{-1, 0, 1\}$. Similarly, we have $a_{12}, \ldots, a_{1n} \in \{-1, 0, 1\}$.

2. Since $A$ is an invertible matrix, the entries $a_{11}, \ldots, a_{1n}$ in the first row of $A$ can not be simultaneously equal to $0$.

3. There is at most only one nonzero number among the entries $a_{11}, \ldots, a_{1n}$. Indeed, assume by the contrary that $a_{11}, a_{12} \neq 0$. Let $x_3 = \ldots = x_n = 0$, from Eq. (60), we have:
$$\sigma(a_{11}x_1 + a_{12}x_2) = a_{11}\sigma(x_1) + a_{12}\sigma(x_2).$$
Note that $a_{11}, a_{12} \in \{-1, 1\}$, so by consider all the cases, we will lead to a contradiction.

It follows from the above three observations that there is exactly one non-zero element among the entries $a_{11}, \ldots, a_{1n}$. In other words, matrix $A$ has exactly one nonzero entry in the first row. This applies for every row, so $A$ has exactly one non-zero entry in each row. Note that, since $A$ is invertible, each column of $A$ has at least one non-zero entry. Therefore, $A$ also has exactly one non-zero entry in each column. Hence, $A$ is in $\mathcal{G}_n$. Moreover, all entries of $A$ are in $\{-1, 0, 1\}$, so $A$ is in $\mathcal{G}_n^{\pm 1}$.

It is straight forward to check that for all $A$ in $\mathcal{G}_n^{\pm 1}$ we have $\sigma(A \cdot \mathbf{x}) = A \cdot \sigma(\mathbf{x})$.

The proposition is then proved completely. $\qquad\square$

## C.2   Proof of Proposition 4.4

*Proof.* For both Fully Connected Neural Networks case and Convolutional Neural Networks case, we consider a network $f$ with three layers, with $n_0, n_1, n_2, n_3$ are number of channels at each layer, and its weight space $\mathcal{U}$. We will show the proof for part $(i)$ where activation $\sigma$ is ReLU, and part $(ii)$ can be proved similarly. For part $(i)$, we prove $f$ to be $G$-invariant on its weight space $\mathcal{U}$, for the group $G$ that is defined by:
$$G = \{\mathrm{id}_{\mathcal{G}_{n_3}}\} \times \mathcal{G}_{n_2}^{>0} \times \mathcal{G}_{n_1}^{>0} \times \{\mathrm{id}_{\mathcal{G}_{n_0}}\} < \mathcal{G}_{n_3} \times \mathcal{G}_{n_2} \times \mathcal{G}_{n_1} \times \mathcal{G}_{n_0} = \mathcal{G}_{\mathcal{U}};$$

**Case 1.** $f$ is a Fully Connected Neural Network with three layers, with $n_0, n_1, n_2, n_3$ are number of channels at each layer as in Eq. 5:

$$f(\mathbf{x} \,;\, U, \sigma) = W^{(3)} \cdot \sigma \left( W^{(2)} \cdot \sigma \left( W^{(1)} \cdot \mathbf{x} + b^{(1)} \right) + b^{(2)} \right) + b^{(3)},$$

**Case 2.** $f$ is a Convolutional Neural Network with three layers, with $n_0, n_1, n_2, n_3$ are number of channels at each layer as in Eq. 8:

$$f(\mathbf{x} \,;\, U, \sigma) = W^{(3)} * \sigma \left( W^{(2)} * \sigma \left( W^{(1)} * \mathbf{x} + b^{(1)} \right) + b^{(2)} \right) + b^{(3)}$$

We have some observations:

**For case 1.** For $W \in \mathbb{R}^{m \times n}, \mathbf{x} \in \mathbb{R}^n$ and $a > 0$, we have:

$$a \cdot \sigma(W \cdot \mathbf{x} + b) = \sigma \left( (aW) \cdot \mathbf{x} + (ab) \right).$$

**For case 2.** For simplicity, we consider $*$ as one-dimentional convolutional operator, and other types of convolutions can be treated similarly. For $W = (w_1, \ldots, w_m) \in \mathbb{R}^m, b \in \mathbb{R}$ and $\mathbf{x} = (x_1, \ldots, x_n) \in \mathbb{R}^n$, we have:

$$W * \mathbf{x} + b = \mathbf{y} = (y_1, \ldots, y_{n-m+1}) \in \mathbb{R}^{n-m+1},$$

where:

$$y_i = \sum_{j=1}^{m} w_j x_{i+j-1} + b.$$

So for $a > 0$, we have:

$$a \cdot \sigma(W * \mathbf{x} + b) = \sigma \left( (aW) * \mathbf{x} + (ab) \right).$$

With these two observations, we can see the proofs for both cases are similar to each other. We will show the proof for case 2, when $f$ is a convolutional neural network since it is not trivial as case 1. Now we have $U = (W, b)$ with:

$$W = \left( W^{(3)}, W^{(2)}, W^{(1)} \right),$$
$$b = \left( b^{(3)}, b^{(2)}, b^{(1)} \right).$$

Let $g$ be an element of $G$:

$$g = \left( \mathrm{id}_{\mathcal{G}_{n_3}}, g^{(2)}, g^{(1)}, \mathrm{id}_{\mathcal{G}_{n_0}} \right),$$

where:

$$g^{(2)} = D^{(2)} \cdot P_{\pi_2} = \mathrm{diag} \left( d_1^{(2)}, \ldots, d_{n_2}^{(2)} \right) \cdot P_{\pi_2} \in \mathcal{G}_{n_2}^{>0},$$
$$g^{(1)} = D^{(1)} \cdot P_{\pi_1} = \mathrm{diag} \left( d_1^{(1)}, \ldots, d_{n_1}^{(1)} \right) \cdot P_{\pi_1} \in \mathcal{G}_{n_1}^{>0}.$$

We compute $gU$:

$$gU = (gW, gb),$$
$$gW = \left( (gW)^{(3)}, (gW)^{(2)}, (gW)^{(1)} \right),$$
$$gb = \left( (gb)^{(3)}, (gb)^{(2)}, (gb)^{(1)} \right).$$

where:

$$(gW)^{(3)}_{jk} = \frac{1}{d^{(2)}_k} \cdot W^{(3)}_{j\pi_2^{-1}(k)},$$

$$(gW)^{(2)}_{jk} = \frac{d^{(2)}_j}{d^{(1)}_k} \cdot W^{(2)}_{\pi_2^{-1}(j)\pi_1^{-1}(k)},$$

$$(gW)^{(1)}_{jk} = \frac{d^{(1)}_j}{1} \cdot W^{(1)}_{\pi_1^{-1}(j)k},$$

and,

$$(gb)^{(3)}_j = b^{(3)}_j,$$

$$(gb)^{(2)}_j = d^{(2)}_j \cdot b^{(2)}_{\pi_2^{-1}(j)},$$

$$(gb)^{(1)}_j = d^{(1)}_j \cdot b^{(1)}_{\pi_1^{-1}(j)}.$$

Now we show that $f(\mathbf{x} \, ; \, U, \sigma) = f(\mathbf{x} \, ; \, gU, \sigma)$ for all $\mathbf{x} = (x_1, \ldots, x_{n_0}) \in \mathbb{R}^{n_0}$. For $1 \leqslant i \leqslant n_3$, we compute the $i$-th entry of $f(\mathbf{x} \, ; \, gU, \sigma)$ as follows:

$$f(\mathbf{x} \, ; \, gU, \sigma)_i$$

$$= \sum_{j_2=1}^{n_2} (gW)^{(3)}_{ij_2} * \sigma \left( \sum_{j_1=1}^{n_1} (gW)^{(2)}_{j_2 j_1} * \right.$$

$$\sigma \left( \sum_{j_0=1}^{n_0} (gW)^{(1)}_{j_1 j_0} * x_{j_0} + (gb)^{(1)}_{j_1} \right) + (gb)^{(2)}_{j_2} \Big) + (gb)^{(3)}_i$$

$$= \sum_{j_2=1}^{n_2} \frac{1}{d^{(2)}_{j_2}} \cdot W^{(3)}_{i\pi_2^{-1}(j_2)} * \sigma \left( \sum_{j_1=1}^{n_1} \frac{d^{(2)}_{j_2}}{d^{(1)}_{j_1}} \cdot W^{(2)}_{\pi_2^{-1}(j_2)\pi_1^{-1}(j_1)} * \right.$$

$$\sigma \left( \sum_{j_0=1}^{n_0} \frac{d^{(1)}_{j_1}}{1} \cdot W^{(1)}_{\pi_1^{-1}(j_1)j_0} * x_{j_0} + d^{(1)}_{j_1} \cdot b^{(1)}_{\pi_1^{-1}(j_1)} \right) + d^{(2)}_{j_2} \cdot b^{(2)}_{\pi_2^{-1}(j_2)} \Big) + b^{(3)}_i$$

$$= \sum_{j_2=1}^{n_2} \frac{1}{d^{(2)}_{j_2}} \cdot W^{(3)}_{i\pi_2^{-1}(j_2)} * \sigma \left( \sum_{j_1=1}^{n_1} \frac{d^{(2)}_{j_2}}{d^{(1)}_{j_1}} \cdot W^{(2)}_{\pi_2^{-1}(j_2)\pi_1^{-1}(j_1)} * \right.$$

$$\sigma \left( d^{(1)}_{j_1} \cdot \left( \sum_{j_0=1}^{n_0} W^{(1)}_{\pi_1^{-1}(j_1)j_0} * x_{j_0} + b^{(1)}_{\pi_1^{-1}(j_1)} \right) \right) + d^{(2)}_{j_2} \cdot b^{(2)}_{\pi_2^{-1}(j_2)} \Big) + b^{(3)}_i$$

$$= \sum_{j_2=1}^{n_2} \frac{1}{d^{(2)}_{j_2}} \cdot W^{(3)}_{i\pi_2^{-1}(j_2)} * \sigma \left( \sum_{j_1=1}^{n_1} \frac{d^{(2)}_{j_2}}{d^{(1)}_{j_1}} \cdot W^{(2)}_{\pi_2^{-1}(j_2)\pi_1^{-1}(j_1)} * \right.$$

$$d^{(1)}_{j_1} \cdot \sigma \left( \sum_{j_0=1}^{n_0} W^{(1)}_{\pi_1^{-1}(j_1)j_0} * x_{j_0} + b^{(1)}_{\pi_1^{-1}(j_1)} \right) + d^{(2)}_{j_2} \cdot b^{(2)}_{\pi_2^{-1}(j_2)} \Big) + b^{(3)}_i$$

$$= \sum_{j_2=1}^{n_2} \frac{1}{d^{(2)}_{j_2}} \cdot W^{(3)}_{i\pi_2^{-1}(j_2)} * \sigma \left( \sum_{j_1=1}^{n_1} d^{(2)}_{j_2} \cdot W^{(2)}_{\pi_2^{-1}(j_2)\pi_1^{-1}(j_1)} * \right.$$

$$\sigma \left( \sum_{j_0=1}^{n_0} W^{(1)}_{\pi_1^{-1}(j_1)j_0} * x_{j_0} + b^{(1)}_{\pi_1^{-1}(j_1)} \right) + d^{(2)}_{j_2} \cdot b^{(2)}_{\pi_2^{-1}(j_2)} \Big) + b^{(3)}_i$$

$$= \sum_{j_2=1}^{n_2} \frac{1}{d_{j_2}^{(2)}} \cdot W_{i\pi_2^{-1}(j_2)}^{(3)} * \sigma \left( d_{j_2}^{(2)} \cdot \left( \sum_{j_1=1}^{n_1} W_{\pi_2^{-1}(j_2)\pi_1^{-1}(j_1)}^{(2)} * \right.\right.$$

$$\left.\left. \sigma \left( \sum_{j_0=1}^{n_0} W_{\pi_1^{-1}(j_1)j_0}^{(1)} * x_{j_0} + b_{\pi_1^{-1}(j_1)}^{(1)} \right) + b_{\pi_2^{-1}(j_2)}^{(2)} \right) \right) + b_i^{(3)}$$

$$= \sum_{j_2=1}^{n_2} \frac{1}{d_{j_2}^{(2)}} \cdot W_{i\pi_2^{-1}(j_2)}^{(3)} \cdot d_{j_2}^{(2)} * \sigma \left( \sum_{j_1=1}^{n_1} W_{\pi_2^{-1}(j_2)\pi_1^{-1}(j_1)}^{(2)} * \right.$$

$$\left. \sigma \left( \sum_{j_0=1}^{n_0} W_{\pi_1^{-1}(j_1)j_0}^{(1)} * x_{j_0} + b_{\pi_1^{-1}(j_1)}^{(1)} \right) + b_{\pi_2^{-1}(j_2)}^{(2)} \right) + b_i^{(3)}$$

$$= \sum_{j_2=1}^{n_2} W_{i\pi_2^{-1}(j_2)}^{(3)} * \sigma \left( \sum_{j_1=1}^{n_1} W_{\pi_2^{-1}(j_2)\pi_1^{-1}(j_1)}^{(2)} * \right.$$

$$\left. \sigma \left( \sum_{j_0=1}^{n_0} W_{\pi_1^{-1}(j_1)j_0}^{(1)} * x_{j_0} + b_{\pi_1^{-1}(j_1)}^{(1)} \right) + b_{\pi_2^{-1}(j_2)}^{(2)} \right) + b_i^{(3)}$$

$$= \sum_{j_2=1}^{n_2} W_{ij_2}^{(3)} * \sigma \left( \sum_{j_1=1}^{n_1} W_{j_2j_1}^{(2)} * \sigma \left( \sum_{j_0=1}^{n_0} W_{j_1j_0}^{(1)} * x_{j_0} + b_{j_1}^{(1)} \right) + b_{j_2}^{(2)} \right) + b_i^{(3)}$$

$$= f(\mathbf{x} \, ; \, U, \sigma)_i.$$

End of proof. □

# D  Additional experimental details

## D.1  Runtime and Memory Consumption

We provide the runtime and memory consumption of Monomial-NFNs and the previous NFNs in Tables 10 and 11 to compare the computational and memory costs in the task of predicting CNN generalization (see Section 6.1). It is observable that our model runs faster and consumes significantly less memory than NP/HNP in [71] and GNN-based method in [35]. This highlights the benefits of parameter savings in Monomial-NFN.

Table 10: Runtime of models.

|  | NP [71] | HNP [71] | GNN [35] | Monomial-NFN (ours) |
|---|---|---|---|---|
| Tanh subset | 35m34s | 29m37s | 4h25m17s | **18m23s** |
| ReLU subset | 36m40s | 30m06s | 4h27m29s | **23m47s** |

Table 11: Memory consumption.

|  | NP [71] | HNP [71] | GNN [35] | Monomial-NFN (ours) |
|---|---|---|---|---|
| Tanh subset | 838MB | 856MB | 6390MB | **582MB** |
| ReLU subset | 838MB | 856MB | 6390MB | **560MB** |

## D.2  Comparison of Monomial-NFNs and GNN-based NFNs

We provide experimental result to compare the efficiency of our model and a permutation equivariant GNN-based NFN [35] in two scenarios below.

1. Training the model on augmented train data and testing with the augmented test data (see Tables 12 and 13).

   Here, we present the experimental results on the original dataset and the results on the augmented dataset. The augmentation levels for the ReLU subset are 1, 2, 3, and 4,

corresponding to augmentation ranges of $[1, 10], [1, 10^2], [1, 10^3], [1, 10^4]$. The augmented dataset for the Tanh subset corresponds to the augmentation range of $[-1, 1]$

Table 12: Predict CNN generalization on ReLU subset (augmented train data)

|  | Original | 1 | 2 | 3 | 4 |
|---|---|---|---|---|---|
| GNN [35] | 0.897 | 0.892 | 0.885 | 0.858 | 0.851 |
| Monomial-NF (ours) | **0.922** | **0.920** | **0.919** | **0.920** | **0.920** |

Table 13: Predict CNN generalization on Tanh subset (augmented train data)

|  | Original | Augmented |
|---|---|---|
| GNN [35] | 0.893 | 0.902 |
| Monomial-NFN (ours) | **0.939** | **0.943** |

The results for GNN exhibit a similar trend as other baselines that do not incorporate the scaling symmetry into their architectures. In contrast, our model has stable performance. A notable observation is that the GNN model uses 5.5M parameters (4 times more than our model), occupies 6000MB of memory, and takes 4 hours to train.

2. Training the model on original train data and testing with the augmented test data (see Tables 14 and 15).

Table 14: Predict CNN generalization on ReLU subset (original train data)

| Augment level | 1 | 2 | 3 | 4 |
|---|---|---|---|---|
| GNN [35] | 0.794 | 0.679 | 0.586 | 0.562 |
| Monomial-NF (ours) | **0.920** | **0.919** | **0.920** | **0.920** |

Table 15: Predict CNN generalization on Tanh subset (original train data)

|  | Augmented |
|---|---|
| GNN [35] | 0.883 |
| Monomial-NFN (ours) | **0.940** |

In these more challenging scenario, GNN's performance drops significantly, which highlights the lack of scaling symmetry in the model. Our model maintains consistent performance, matching the case in which we train with the augmented data.

### D.3 Predicting generalization from weights

**Dataset.** The original $\mathrm{ReLU}$ subset of the CNN Zoo dataset includes 6050 instances for training and 1513 instances for testing. For the $\mathrm{Tanh}$ dataset, it includes 5949 training and 1488 testing instances. For the augmented data, we set the augmentation factor to 2, which means that we augment the original data once, resulting in a new dataset of double the size. The complete size of all datasets is presented in Table 16

**Implementation details.** Our model follows the same architecture as in [71], comprising three equivariant Monomial-NFN layers with 16, 16, and 5 channels, respectively, each followed by $\mathrm{ReLU}$ activation (ReLU dataset) or $\mathrm{Tanh}$ activation (Tanh dataset). The resulting weight space features are input into an invariant Monomial-NFN layer with Monomial-NFN pooling (Equation 19) with learnable parameters (ReLU case) or mean pooling (Tanh case). Specifically, the Monomial-NFN pooling layer normalizes the weights across the hidden dimension and takes the average for rows (first layer), columns (last layer), or both (other layers). The output of this invariant Monomial-NFN layer is flattened and projected to $\mathbb{R}^{200}$ (ReLU case) or $\mathbb{R}^{1000}$ (Tanh case). This resulting vector is then passed through an MLP with two hidden layers with $\mathrm{ReLU}$ activations. The output is linearly projected to a scalar and then passed through a sigmoid function. We use the Binary Cross Entropy (BCE) loss function and train the model for 50 epochs, with early stopping based on $\tau$ on the validation set, which takes 35 minutes to train on an A100 GPU. The hyperparameters for our model are presented in Table 18.

Table 16: Datasets information for predicting generalization task.

| Dataset | Train size | Val size |
|---|---|---|
| Original ReLU | 6050 | 1513 |
| Original Tanh | 5949 | 1488 |
| Augment ReLU | 12100 | 3026 |
| Augment Tanh | 11898 | 2976 |

Table 17: Number of parameters of all models for prediciting generalization task.

| Model | ReLU dataset | Tanh dataset |
|---|---|---|
| STATNN | 1.06M | 1.06M |
| NP | 2.03M | 2.03M |
| HNP | 2.81M | 2.81M |
| Monomial-NFN (ours) | 0.25M | 1.41M |

Table 18: Hyperparameters for Monomial-NFN on prediciting generalization task.

| | ReLU | Tanh |
|---|---|---|
| MLP hidden neurons | 200 | 1000 |
| Loss | Binary cross-entropy | Binary cross-entropy |
| Optimizer | Adam | Adam |
| Learning rate | 0.001 | 0.001 |
| Batch size | 8 | 8 |
| Epoch | 50 | 50 |

Table 19: Dataset size for Classifying INRs task.

| | Train | Validation | Test |
|---|---|---|---|
| CIFAR-10 | 45000 | 5000 | 10000 |
| MNIST size | 45000 | 5000 | 10000 |
| Fashion-MNIST | 45000 | 5000 | 20000 |

For the baseline models, we follow the original implementations described in [71], using the official code (available at: https://github.com/AllanYangZhou/nfn). For the HNP and NP models, there are 3 equivariant layers with 16, 16, and 5 channels, respectively. The features go through an average pooling layer and 3 MLP layers with 1000 hidden neurons. The hyperparameters of our model and the number of parameters for all models in this task can be found in Table 17.

### D.4    Classifying implicit neural representations of images

**Dataset.**    We utilize the original INRs dataset provided by [71], with no augmentation. The data is obtained by implementing a single SIREN model for each image in each dataset: CIFAR-10, MNIST, and Fashion-MNIST. The size of training, validation, and test samples for each dataset is provided in Table 19.

**Implementation details.**    In these experiments, our general architecture includes 2 Monomial-NFN layers with sine activation, followed by 1 Monomial-NFN layer with absolute activation. The choice of hidden dimension in the Monomial-NFN layer depends on each dataset and is described in Table 20. The architecture then follows the same design as the NP and HNP models in [71], where a Gaussian Fourier Transformation is applied to encode the input with sine and cosine components, mapping from 1 dimension to 256 dimensions. If the base layer is NP, the features will go through IOSinusoidalEncoding, a positional encoding designed for the NP layer, with a maximum frequency of 10 and 6 frequency bands. After that, the features go through 3 HNP or NP layers with ReLU activation functions. Then, an average pooling is applied, and the output is flattened, and the resulting vector is passed through an MLP with two hidden layers, each containing 1000 units and ReLU activations. Finally, the output is linearly projected to a scalar. For the MNIST dataset, there is an additional Channel Dropout layer after the ReLU activation of each HNP layer and a Dropout layer after the ReLU activation of each MLP layer, both with a dropout rate of 0.1. We use the Binary Cross Entropy (BCE) loss function and train the model for 200,000 steps, which takes 1 hour and 35

Table 20: Hyperparameters of Monomial-NFN for each dataset in Classify INRs task.

| | MNIST | Fashion-MNIST | CIFAR-10 |
|---|---|---|---|
| Monomial-NFN hidden dimension | 64 | 64 | 16 |
| Base model | HNP | NP | HNP |
| Base model hidden dimension | 256 | 256 | 256 |
| MLP hidden neurons | 1000 | 500 | 1000 |
| Dropout | 0.1 | 0 | 0 |
| Learning rate | 0.000075 | 0.0001 | 0.0001 |
| Batch size | 32 | 32 | 32 |
| Step | 200000 | 200000 | 200000 |
| Loss | Binary cross-entropy | Binary cross-entropy | Binary cross-entropy |

Table 21: Number of parameters of all models for classifying INRs task.

| | CIFAR-10 | MNIST | Fashion-MNIST |
|---|---|---|---|
| MLP | 2M | 2M | 2M |
| NP | 16M | 15M | 15M |
| HNP | 42M | 22M | 22M |
| Monomial-NFN (ours) | 16M | 22M | 20M |

Table 22: Number of parameters of all models for Weight space style editing task.

| Model | Number of parameters |
|---|---|
| MLP | 4.5M |
| NP | 4.1M |
| HNP | 12.8M |
| Monomial-NFN (ours) | 4.1M |

Table 23: Hyperparameters for Monomial-NFN on weight space style editing task.

| Name | Value |
|---|---|
| Monomial-NFN hidden dimension | 16 |
| NP dimension | 128 |
| Optimizer | Adam |
| Learning rate | 0.001 |
| Batch size | 32 |
| Steps | 50000 |

minutes on an A100 GPU. For the baseline models, we follow the same architecture in [71], with minor modifications to the model hidden dimension, reducing it from 512 to 256 to avoid overfitting. We use a hidden dimension of 256 for all baseline models and our base model. The number of parameters of all models can be found in Table 21

### D.5 Weight space style editing

**Dataset.** We use the same INRs dataset as used for classification task, which has the size of train, validation and test set described in Table 19.

**Implementation details.** In these experiments, our general architecture includes 2 Monomial-NFN layers with 16 hidden dimensions. The architecture then follows the same design as the NP model in [71], where a Gaussian Fourier Transformation with a mapping size of 256 is applied. After that, the features go through IOSinusoidalEncoding and then through 3 NP layers, each with 128 hidden dimensions and $\mathrm{ReLU}$ activation. Finally, the output goes through an NP layer to project into a scalar and a LearnedScale layer described in the Appendix of [71]. We use the Binary Cross Entropy (BCE) loss function and train the model for 50,000 steps, which takes 35 minutes on an A100 GPU. For the baseline models, we keep the same settings as the official implementation. Specifically, the HNP or NP model will have 3 layers, each with 128 hidden dimensions, followed by a $\mathrm{ReLU}$ activation. An NFN of the same type will be applied to map the output to 1 dimension and pass it through a LearnedScale layer. The number of parameters of all models can be found in Table 22. The detailed hyperparameters for our model can be found in Table 23.

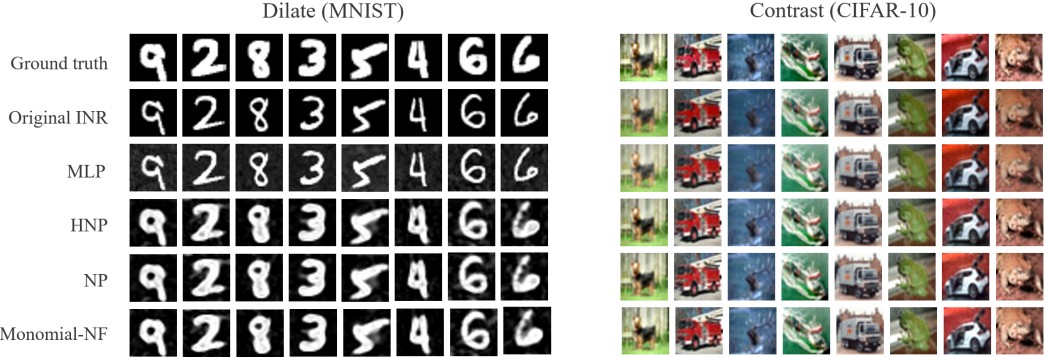

Figure 2: Random qualitative samples of INR editing behavior on the Dilate (MNIST) and Contrast (CIFAR-10) editing tasks.

## D.6 Ablation Regarding Design Choices

We provide the ablation study on the choice of architecture for the task Predict CNN Generalization on ReLU subset in Table 24. We denote:

- Monomial Equivariant Functional Layer (Ours): MNF
- Activation: ReLU
- Scaling Invariant and Permutation Equivariant Layer (Ours): Norm
- Hidden Neuron Permutation Invariant Layer (in [71]): HNP
- Permutation Invariant Layer: Avg
- Multilayer Perceptron: MLP

Table 24: Ablation study on design choices for the task Predict CNN generalization on ReLU subset

| | Original | 1 | 2 | 3 | 4 |
|---|---|---|---|---|---|
| (MNF–ReLU)×1 → Norm → (HNP–ReLU)×1 → Avg → MLP | 0.917 | 0.916 | 0.917 | 0.917 | 0.917 |
| (MNF–ReLU)×2 → Norm → (HNP–ReLU)×1 → Avg → MLP | 0.918 | 0.917 | 0.917 | 0.917 | 0.918 |
| (MNF–ReLU)×3 → Norm → (HNP–ReLU)×1 → Avg → MLP | 0.920 | 0.919 | 0.918 | 0.920 | 0.920 |
| (MNF–ReLU)×1 → Norm → Avg → MLP | 0.915 | 0.914 | 0.917 | 0.916 | 0.914 |
| (MNF–ReLU)×2 → Norm → Avg → MLP | 0.918 | 0.919 | 0.918 | 0.917 | 0.918 |
| (MNF–ReLU)×3 → Norm → Avg → MLP | **0.922** | **0.920** | **0.919** | **0.920** | **0.920** |

Among these designs, the architecture incorporating three layers of Monomial-NFN with ReLU activation achieves the best performance.

