# OpenReview forum: "Monomial Matrix Group Equivariant Neural Functional Networks"
_NeurIPS.cc/2024/Conference — NeurIPS 2024 poster_

### Official Review · Reviewer_xQCm · 2024-07-05

**Soundness:** 2
**Presentation:** 3
**Contribution:** 2
**Rating:** 5
**Confidence:** 4

**Summary:**

The paper explores the important field of learning over weight spaces, where neural networks process other neural networks. Previous research has highlighted the importance of designing equivariant architectures that account for the symmetries of the input neural network, with a primary focus on permutation symmetries. However, previous literature did not account for all symmetries of the input NNs, particularly the weight scaling symmetries of ReLU networks and the weight sign flipping symmetries of sin or Tanh networks. This paper naturally extends previous approaches to account for these activation-based symmetries.
This paper first formalizes the group of symmetries that includes both neuron permutations and scaling or sign-flipping transformations using monomial matrices. Next, it proposes a novel architecture for weight space networks that are equivariant to groups of monomial matrices. The new architecture is more efficient in terms of the number of trainable parameters compared to baseline weight space networks (which are only permutation equivariant).

**Strengths:**

1. The paper is mostly well-written and well-structured.
2. The paper addresses the important and timely problem of learning in deep weight spaces, presenting a novel architecture that extends previous permutation equivariant networks to also account for scale/sign-flipping symmetries.
3. The proposed architecture is parameter efficient compared to baseline weight space networks.

**Weaknesses:**

My main concern is the limited empirical evaluation and missing natural baselines. Also, the presented empirical results, mostly show marginal to no improvement over the limited baseline methods evaluated. The insufficient empirical study of this paper significantly damages the paper’s contribution. Given the current state of the learning over weight spaces literature, I would expect a more diverse, challenging, and comprehensive empirical evaluation.

1. The main text provides very few details on the construction of G-equivariant layers. I suggest the authors to provide at least one concrete example for mapping between a subspace of U, for example, mapping between some bias $b_i$ to a bias $b’_j$.
2. Since the method is built over NFN, it is limited in the sense that each monomial-NFN can process a specific input architecture. Building over or extending the work to GNN-based weight space networks will allow [1,2] the processing of diverse input architectures.
3. Missing weight-space baselines, like GNN-based models [1,2], DWSNets [3] (which, while mathematically equiv. to NFNs, obtains better empirical performance, see e.g., [6]) and NFT [4].
4. Another important missing natural baseline is to use a permutation equivariant baseline like DWS/NFN or the GNN-based models together with scaling/sign-flipping data augmentations as in [5].
5. Some evaluation and comparison of runtime and memory consumption w.r.t. baselines would be beneficial.
6. Also, adding some ablation regarding design choices would be beneficial.

References

[1] Graph neural networks for learning equivariant representations of neural networks, ICLR 2024.

[2] Graph metanetworks for processing diverse neural architectures, ICLR 2024.

[3] Equivariant Architectures for Learning in Deep Weight Spaces, ICML 2023.

[4] Neural Functional Transformers, NeurIPS 2024.

[5] Improved Generalization of Weight Space Networks via Augmentations, ICML 2024.

[6] Neural Processing of Tri-Plane Hybrid Neural Fields, ICLR 2024.

**Questions:**

1. Is it possible to use the proposed method together with normalization layers?
2. How easy would it be to extend the method to other activation functions?

**Limitations:**

Yes.

---

> ### Author Rebuttal · Authors · 2024-08-06
>
> Thank you for your thoughtful review and valuable feedback. Below we address your concerns.
>
> **W1: Concrete example.**
>
> **Answer:** See an illustrative example in **A4** of General Response.
>
> **W2: Each Monomial-NFN can process a specific input architecture. Building over or extending the work to GNN-based weight space networks will allow [D1,D2] the processing of diverse input architectures.**
>
> **Answer:** See **Q1-Q2** in General response. In addition, while [D1,D2] can handle diverse input architectures, we do not know whether it is possible to incorporate non-permutations, such as scaling and sign-flipping symmetries, into these models.
>
> In contrast, we take the first step toward incorporating non-permutation symmetries into NFNs. In particular, our proposed model is equivariant to permutations and scaling (for ReLU networks) and sign-flipping (for sin and tanh networks). This leads to a significant reduction in the number of parameters, which is particularly useful for large NNs in modern deep learning, while achieving comparable or better results than those in the literature.
>
>
>
> **W3: Missing weight-space baselines, like GNN-based models [D1,D2], DWSNets [D3] (which, while mathematically equiv. to NFNs, obtains better empirical performance, see e.g., [D6]) and NFT [D4]**
>
> **W4: Another important missing natural baseline is to use a permutation equivariant baseline like DWS/NFN or the GNN-based models together with scaling/sign-flipping data augmentations as in [D5].**
>
> **Answer to W3-W4:**
> Thank you for pointing out these related works. Here we provide the experimental results for GNN [D1] in two scenarios:
> 1. Training the model on augmented train data and test with the augmented test data
>
> *Table 1: Predict CNN generalization on ReLU subset (augmented train data)*
> | |Original |1|2|3|4|
> |-|:-:|:-:|:-:|:-:|:-:|
> | GNN [D1]|0.897|0.892|0.885|0.858|0.851|
> | Monomial-NF (ours) | **0.922** | **0.920** | **0.919** | **0.920** | **0.920** |
>
> *Table 2: Predict CNN generalization on Tanh subset (augmented train data)*
>
> || Original| Augmented |
> |-|:-:|:-:|
> | GNN [D1]| 0.893| 0.902|
> | Monomial-NFN (ours) | **0.939** | **0.943** |
>
> The results for GNN exhibit a similar trend as other baselines that do not incoporate the scaling symmetry into their architectures. In contrast, our model has stable performance. A notable observation is that GNN model uses 5.5M parameters (4 times more than our model), occupies 6000MB of memory and takes 4 hours to train (refer to **Q3** in General Response).
>
> 2. Train the model on original train data and test with the augmented test data
>
> *Table 3: Predict CNN generalization on ReLU subset (original train data)*
> |Augment level|1|2|3|4|
> |-|:-:|:-:|:-:|:-:|
> | GNN [D1]|0.794| 0.679|0.586 |0.562|
> | Monomial-NF (ours) |  **0.920** | **0.919** | **0.920** | **0.920** |
>
> *Table 4: Predict CNN generalization on Tanh subset (original train data)*
> | | Augmented |
> |-|:-:|
> | GNN [D1]| 0.883|
> | Monomial-NFN (ours) | **0.940** |
>
>
> In this more challenging scenario, GNN's performance drops significantly, which highlights the lack of scaling symmetry in the model. Our model maintains consistent performance, matching the case in which we train with the augmented data.
>
> **W5: Comparison of runtime and memory consumption w.r.t. baselines.**
>
> **Answer:** See **Q3** in General Response.
>
>
> **W6: Ablation regarding design choices.**
>
> **Answer:**
>
> Here we provide the ablation study on the choice of architecture for the task Predict CNN Generalization on ReLU subset. We denote:
>
> - Monomial Equivariant Functional Layer (Ours): MNF
> - Activation: ReLU
> - Scaling Invariant and Permutation Equivariant Layer (Ours): Norm
> - Hidden Neuron Permutation Invariant Layer (in [D7]): HNP
> - Permutation Invariant Layer: Avg
> - Multilayer Perceptron: MLP
>
> *Table 5: Ablation study on design choices for the task Predict CNN generalization on ReLU subset*
> | |Original|1|2|3|4|
> |-|:-:|:-:|:-:|:-:|:-:|
> |(MNF → ReLU)x1 → Norm → (HNP → ReLU)x1 → Avg → MLP |0.917|0.916|0.917|0.917|0.917|
> | (MNF → ReLU)x2 → Norm → (HNP → ReLU)x1 → Avg → MLP |0.918|0.917|0.917|0.917|0.918|
> | (MNF → ReLU)x3 → Norm → (HNP → ReLU)x1 → Avg → MLP |0.920|0.919|0.918|0.920|0.920|
> | (MNF → ReLU)x1 → Norm → Avg → MLP |0.915|   0.914|0.917|0.916|0.914|
> | (MNF → ReLU)x2 → Norm → Avg → MLP |0.918|0.919|0.918|0.917|0.918|
> | (MNF → ReLU)x3 → Norm → Avg → MLP | **0.922** | **0.920** | **0.919** | **0.920** | **0.920** |
>
> Among these designs, the architecture incoporating three layers of Monomial-NFN with ReLU activation achieves the best performance.
>
>
> **Q1: Is it possible to use the proposed method together with normalization layers?**
>
> **Answer:** See **Q2** in General Response.
>
> **Q2: How easy would it be to extend the method to other activation functions?**
>
> **Answer:** See **Q1** in General Response.
>
>
> **References**
>
> [D1] Graph neural networks for learning equivariant representations of neural networks, ICLR 2024.
>
> [D2] Graph metanetworks for processing diverse neural architectures, ICLR 2024.
>
> [D3] Equivariant Architectures for Learning in Deep Weight Spaces, ICML 2023.
>
> [D4] Neural Functional Transformers, NeurIPS 2023.
>
> [D5] Improved Generalization of Weight Space Networks via Augmentations, ICML 2024.
>
> [D6] Neural Processing of Tri-Plane Hybrid Neural Fields, ICLR 2024.
>
> [D7] Permutation Equivariant Neural Functionals, NeurIPS 2023

---

> ### Author Response · Authors · 2024-08-10
> **Any Questions from Reviewer xQCm on Our Rebuttal?**
>
> We would like to thank the reviewer again for your thoughtful reviews and valuable feedback.
>
> We would appreciate it if you could let us know if our responses have addressed your concerns and whether you still have any other questions about our rebuttal.
>
> We would be happy to do any follow-up discussion or address any additional comments.

---

> ### Comment · Reviewer_xQCm · 2024-08-11
> **Response to Rebuttal**
>
> I would like to thank the authors for their rebuttal and for providing additional results and discussion, which addresses some of my concerns. I've read all the reviewers' concerns and the authors' responses. While not all of my main concerns were addressed, I do appreciate and acknowledge the novelty and contribution of the paper, and I will raise my score to align with the accepted score of all reviewers.
> I encourage the authors to include additional baselines and more strong evaluation comparisons in the revised version of the paper.

---

> > ### Author Response · Authors · 2024-08-12
> > **Thanks for your endorsement!**
> >
> > Thanks for your reply, and we appreciate your endorsement. Following your suggestion, we will include additional baselines and stronger evaluation comparisons in our revision.

---

### Official Review · Reviewer_yJwm · 2024-07-12

**Soundness:** 4
**Presentation:** 4
**Contribution:** 3
**Rating:** 7
**Confidence:** 5

**Summary:**

The present manuscript concerns the design of Neural Networks capable of processing the weights and biases of other Neural Networks, particularly Fully Connected and Convolutional NNs (known in the literature as *weight space networks, neural functional networks or metanetworks*). Previous works considered only hidden-neuron-permutation symmetries of weights/biases that preserve the function of the NN. In contrast, the authors highlight the existence of symmetries induced by certain activation functions (ReLU – positive scaling symmetries and sine/tanh – sign symmetries – Eq. (20)), following the works of Godfrey et al., NeurIPS’22 and Wood and Shawe-Taylor, Discr. Appl. Math.’96, where in certain cases are mentioned to be maximal (i.e. the only symmetries that preserve the NN function) as proved in the past. Combining these symmetries with permutations leads to the so-called “monomial-matrix groups”. Based on this background, the authors design monomial-matrix equivariant NNs, following the classical NN design strategy: linear layers interleaved with non-linearities. To characterise the former, they identify the weight-sharing pattern that equivariance imposes by solving a system of weight constraints for both the symmetries above (thus fully characterising linear layers), while for the former they use the same non-linearities as the activation functions at hand. Finally, they propose a method for monomial-matrix *invariant* layer design, which is combined with equivariant layers to yield an end-to-end invariant NN. The method is experimentally tested in various tasks: CNN generalisation prediction and Implicit Neural Representation (INR) classification and editing, showing competitive performance against permutation-symmetry-only baselines and a significant reduction in the number of learnable parameters.

**Strengths:**

**Significance/Impact.** The topic of NN processing has been steadily gaining traction in the last year and has the potential to provide significant advantages in various applications such as meta-learning and processing signals of arbitrary nature (encoded into INRs) under a unifying framework. Therefore, improving computational efficiency and incorporating new inductive biases, as done in this work, is an important step towards advancing and popularising the field.

**Presentation**. The paper is in general well-presented, with appropriately chosen notations and clear descriptions of the background concepts involved and the innovations proposed.

**Novelty**.
   - *Studied problem*. The symmetries discussed in this work, although mentioned in the literature, have not been approached so far in the context of weight space networks.
   - *Methodology*: This work introduces novel layers that are equivariant or invariant to a group that has been underexplored (monomial matrices).
   - *Theory*. Additionally, following traditional weight-sharing proof strategies, the layers are characterized as the only linear layers that have the equivariance property, while Remark 4.5. borrows results from relevant papers (that might be not widely known to the community) to highlight the cases where the studied symmetries are the maximally function-preserving ones (however some results are missing – see weaknesses).

**Quality/Execution**. The paper is well-motivated, provides a comprehensive background discussion, follows a rigorous and well-established methodology to design monomial-matrix group equivariant/invariant layers and provides adequate experimental comparisons, including in regimes where previous works fail.

**Computational efficiency**. The proposed method leads to a significant reduction in the number of parameters, a property which is particularly useful for large NNs (a typical use case in modern deep learning).

**Weaknesses:**

**Limited expressivity (possibly reflected in the experimental results?)**
- Although characterising the linear layers by solving the weight sharing constraints is a fairly general and rigorous technique for equivariant layer design, I have the impression that the resulting weight sharing, in this case, is severe. I.e. due to the large size of the group considered, the resulting layers seem weak in terms of expressivity. For example, in Eq. (22) all hidden layer weights and bias updates are calculated by linearly processing each element individually, i.e. a weight corresponding to an edge between two neurons will be updated based only on its previous value, ignoring other edges across the same or other layers.
- Additionally, the activation functions that can be used are quite limited to preserve equivariance (the authors mention that they use the same activation as in the NN that is being processed).
- Although these design choices are necessary for the current construction, this probably will not be the case for a different one, i.e. by directly designing non-linear layers. In other words, I am concerned that working with the standard NN pattern (linear layers interleaved with non-linearities) might be too limiting for this family of symmetries.
- I am also wondering if limitations in expressivity are induced by the choice of the invariant layers. Could the authors elaborate on this?
- I do not consider the above as grounds for rejection, since I believe that even incorporating these symmetries into NFNs and the linear layer characterisation are sufficient contributions. However, they seem like important limitations, which I suspect are reflected in the experimental results, and therefore should be highlighted by the authors.

**Related work and existing results**.
- Given that the field is relatively new, I think that the authors should devote more space to a more detailed literature review, e.g. describing the weight symmetries that have been discovered more thoroughly and discussing/comparing the weight space networks that have been proposed so far. Currently, most methods are cited, but not adequately explained. I understand that space might not allow this, but at least adding an extended related work section in the appendix would help.
- Additionally, the following very related works are missing:  (1) In the topic of weight space networks:
   - Universal Neural Functionals, Zhou et al., arxiv’24
   - Graph metanetworks for processing diverse neural architectures, Lim et al., ICLR’23
- (2) In the topic of weight space symmetries (conditions for maximality):
   - Reverse-engineering deep ReLU networks, Rolnick et al., ICML’20
   - Hidden Symmetries of ReLU Networks, Grigsby et al., ICML’23
- As far as I understand, the works of Chen et al., Neur. Comp.’93, Fefferman et al., NIPS’93 characterise maximal weight space symmetries only for the tanh activation and not for sine, as the authors mention L208 – in case this is true, the statement in Remark 4.5. should be rectified.

**Extensibility**. Judging by the derivation of the weight-sharing patterns, it appears that it is not straightforward to extend this approach to symmetries induced by other activation functions (e.g. some are discussed in Godfrey et al.). Could the authors discuss this (and if true include it in their discussion about limitations?).

**Questions:**

- As far as I know, Proposition 3.4. and Proposition 4.4 follow from the characterisation of intertwiner groups done by Godfrey et al., NeurIPS’22 (e.g. see Theorem E.14 and Proposition 3.4), but the authors provide alternative proofs. In case these proofs are of independent technical interest, I would recommend that the authors briefly mention a proof sketch in the main paper and the differences with Godfrey et al. (this would also help in properly accrediting this prior work).
- L210: The authors mention “It is natural to ask whether the group $G$ is still maximal in the other case”. What do they mean by the “other case”?
- Perhaps an illustrative example of the resulting weight-sharing would help. Or maybe give some intuition on the resulting layers?
- Maybe adding the exact number of parameters along with the performance in the tables would help in grasping the actual reduction.

**Minor**.
- L126-L128 are unclear. Perhaps describe in more detail the notation $\text{Aut}(\Delta_n)$ and the terms conjugation, semi-direct product etc. (depending on the importance of these statements).
- L242: hyperparams --> Perhaps you mean learnable params?

**Limitations:**

Some of the limitations of this work are mentioned in the text (last paragraph of Section 5 and conclusion section). However, I think that some have not been adequately discussed (see weaknesses). I would recommend adding a separate section for this purpose.

I do not foresee any negative societal impact.

---

> ### Author Rebuttal · Authors · 2024-08-06
>
> Thank you for your thoughtful review and valuable feedback. Below we address your concerns.
>
> -----
>
>
> **W1: Limited Expressivity.**
>
> **Answer:** We agree with the reviewer's discussion on limitations. However, we would also like to share our thoughts on these limitations.
>
> - Although **large size symmetry group might lead to the small number of independent parameters**, we believe that the resulting equivariant layers are still sufficient in terms of expressivity (see Theorem 5.1). Nevertherless, it is necessary to construct equivariant nonlinear layers that can encode more relations between weight network's parameters in order to achieve better results. We leave this interesting topic for future study.
> - **About the activation functions can be used**: See **Q1** in General Response.
> - **Beyond the linear layers**, the problem of characterizing nonlinear layers which are equivariant to both permutations and scaling/sign-flipping, or in particular, the problem of incorporating scaling/sign-flipping symmetries into nonlinear layers (such as self-attentions) is a nontrivial problem. We leave this open problem for future study.
> - **About the choice of the invariant layers**: Unfortunately, the invariant layer constructed in our paper is not expressive enough to express all invariant NFNs. For the ReLU case, one can verify that every invariant layer can be expressed via positively homogenenous functions of degree zero via Eq. (55). However, not all positively homogeneous functions of degree zero can be written in the form of the candidate choice in Eq. (56). Nevertheless, our candidate choice in Eq. (56) already covers a large part of positively homogeneous functions of degree zero. As a result, the invariant layers yield favorable experimental results. The same arguments apply for the sine and tanh networks. We have added this discussion to the limitations.
>
> **W2: Related work and existing results.**
>
> **Answer:** Following the reviewer's suggestion, we have added an extended related work section in the appendix to: (1) include missing related works in the topic of weight space networks and  weight space symmetries, and (2) provide more adequate explanations of cited methods. We have also editted Remark 4.5 regarding the works of Chen et al., Neur. Comp.’93, Fefferman et al., NIPS’93 on characterising maximal weight space symmetries only for the tanh activation and not for sine.
>
> **W3: Extensibility of this approach to other activation functions.**
>
> **Answer:** See **Q1** in General Response.
>
> **Q1: About the proofs of Propositions 3.4. and 4.4.**
>
> **Answer:** While the proofs in Godfrey et al. (NeurIPS '22) are technically involved and apply to very general cases, we provide direct and simple proofs that apply to our considered cases for the convenience of the reader and the completeness of the paper. Our proofs can be seen as simplified versions of those in Godfrey et al. (NeurIPS '22), justified for the considered cases.
>
> **Q2: What do they mean by the "other case"?**
>
> **Answer:** By the "other case", we mean the case when the network architectures are MLPs with ReLU activation such that the condition $n_L \geq \ldots \geq n_2 \geq n_1 > n_0 =1$ is not satisfied. In addition, regarding the works of [Chen et al., Neur. Comp.’93], [Fefferman et al., NIPS’93] on characterising maximal weight space symmetries only for the tanh activation and not for sine, the "other case" now contains MLPs with sin activation, too. We have added this discussion to the revised version.
>
> **Q3: An illustrative example.**
>
> **Answer:** See **Q4** in General Response.
>
> **Q4: Adding the exact number of parameters along with the performance.**
>
> **Answer:** The exact number of parameters for all models in all tasks have been provided in the Appendix. In addition, we have added runtime and memory usage of our model and the previous ones (see **Q3** in General Response).

---

> ### Author Response · Authors · 2024-08-10
> **Any Questions from Reviewer yJwm on Our Rebuttal?**
>
> We would like to thank the reviewer again for your thoughtful reviews and valuable feedback.
>
> We would appreciate it if you could let us know if our responses have addressed your concerns and whether you still have any other questions about our rebuttal.
>
> We would be happy to do any follow-up discussion or address any additional comments.

---

> > ### Comment · Reviewer_yJwm · 2024-08-14
> > **Post rebuttal**
> >
> > I thank the authors for their response. As I mentioned in my initial review, I do not have strong objections to this paper, apart from the fact that expressivity could be limited (we currently do not know, but there are some hints). My overall assessment of the paper remains the same and I recommend acceptance.
> >
> > **Suggestion to authors:** However, I think the discussion on expressivity could have been more elaborate. For example, the argument that the "layers are still sufficient in terms of expressivity (see Theorem 5.1)." is not convincing since Theorem 5.1. only characterises linear equivariant layers and does not provide any evidence about non-linear functions. Another remaining question is whether expressivity is affected because of the need to use only certain activation functions in the neural functional/metanetwork - this has not been thoroughly discussed (the authors pointed to their general response, but I could not locate this part). I strongly encourage the authors to be upfront about the above in their limitations/open questions section to make their work more complete.

---

> > > ### Author Response · Authors · 2024-08-14
> > >
> > > Thank you very much for your reply. We agree with your suggestion, and we will include these interesting discussions, such as the expressivity of the equivariant/invariant layers and the effects of activations, in the limitations and open questions section for future work.

---

### Official Review · Reviewer_MDVG · 2024-07-12

**Soundness:** 4
**Presentation:** 3
**Contribution:** 3
**Rating:** 7
**Confidence:** 3

**Summary:**

This paper studies the extension of permutation equivariant neural functionals to accommodate the monomial group, which is a generalization of the permutation group. This extension leads to a new class of NFN called monomial NFN that can also handle the scaling symmetry of positively homogenous activation (RELU) and sign-flipping symmetry of activations such as tanh. The paper then characterizes the subset of all monomial matrices that are either *preserved* by either ReLU or sin/tanh ($\sigma A(x) = A(\sigma x)$). The paper then proceeds to construct group $G_\mathcal{U}$ which is the product of all monomial groups that act on a weight space object $U$, and two subgroups of $G_\mathcal{U}$ under which the models with ReLU or sin / tanh are invariant. Finally, the paper constructs an affine weight space layer that is $G$-equivariant and also $G$-invariant weight space layers, which are more parameter efficient than prior works. Generalization prediction experiments show that the proposed layer performs better than prior works under standard conditions and significantly outperforms when the models are perturbed by scale, corroborating the statement made by the paper. On INR classification and editing tasks, monomial-NFNs are competitive or better than prior works.

**Strengths:**

Overall, this is a well-written and technically solid paper with a novel contribution to the neural functional literature. The theoretical results seem sound. While I did not thoroughly check every proof detail, Figure 1 demonstrates that the constructed layer is indeed equivariant to the scale of the weights. The other experimental results are also quite compelling. The parameter saving could also potentially be a big upside of the monomial-NFN which can bring better generalization.

**Weaknesses:**

I don't see major weaknesses in the paper. Perhaps one shortcoming is that the majority of performance improvement comes when the weights are scaled, which is not a very natural perturbation in reality. On the other hand, many tasks that cannot be "solved" by previous NFNs remain difficult for monomial NFNs (e.g., CIFAR10 classification), which might indicate some fundamental limitation of this line of work.

**Questions:**

1. How do the computation cost and memory compare to NFNs? In my opinion, the issue with NFNs' big parameter count is that it's very memory-intensive. While monomial-NFN saves parameters, the benefit is perhaps less significant if it doesn't come with a computational advantage.

**Limitations:**

Authors have adequately discussed the limitation though it would be good to include some computation cost analysis.

---

> ### Author Rebuttal · Authors · 2024-08-06
>
> Thank you for your thoughtful review and valuable feedback. Below we address your concerns.
>
> **Weakness: I don't see major weaknesses in the paper. Perhaps one shortcoming is that the majority of performance improvement comes when the weights are scaled, which is not a very natural perturbation in reality. On the other hand, many tasks that cannot be "solved" by previous NFNs remain difficult for monomial NFNs (e.g., CIFAR10 classification), which might indicate some fundamental limitation of this line of work.**
>
> **Answer:** We agree with the reviewer on these limitations of this line of work. In addition, the problem of constructing different types of architecture, such as GNN-based or self-attention-based models, that are equivariant to a monomial matrix group is interesting and nontrivial. We leave this open problem for future study.
>
> **Q1: How do the computation cost and memory compare to NFNs? In my opinion, the issue with NFNs' big parameter count is that it's very memory-intensive. While monomial-NFN saves parameters, the benefit is perhaps less significant if it doesn't come with a computational advantage.**
>
> **Answer:** See **Q3** in General Response.

---

> ### Author Response · Authors · 2024-08-10
> **Any Questions from Reviewer MDVG on Our Rebuttal?**
>
> We would like to thank the reviewer again for your thoughtful reviews and valuable feedback.
>
> We would appreciate it if you could let us know if our responses have addressed your concerns and whether you still have any other questions about our rebuttal.
>
> We would be happy to do any follow-up discussion or address any additional comments.

---

> > ### Comment · Reviewer_MDVG · 2024-08-12
> >
> > Thank you for the response. My opinion of the paper is still positive and I would like to keep my current score.

---

> > > ### Author Response · Authors · 2024-08-13
> > > **Thanks for your endorsement!**
> > >
> > > Thanks for your reply, and we appreciate your endorsement.

---

### Official Review · Reviewer_RAv9 · 2024-07-13

**Soundness:** 3
**Presentation:** 3
**Contribution:** 3
**Rating:** 5
**Confidence:** 5

**Summary:**

- Paper improves neural functional networks by taking into account weight scaling properties of ReLU networks and weight sign flipping symmetries of sin or Tanh networks.

- Monomial matrices are used to represent these symmetries, both permutation and scale/sign-flipping transformations.

- Proposed model has fewer trainable (independent) parameters compared to the original NFN family of models.

**Strengths:**

- Proposes a principled way to incorporate activation functions in neural network representations for MLPs and CNNs.

**Weaknesses:**

- No discussion/comparison with previous works on the subject [1,2,3,5].

- No discussion of extensions to architectures with branches/ transformers?

- Not the first to consider activation functions in representing neural network weights as claimed. In [1] and [2], activation functions are encoded as nodes in a graph.

- The performance gain over the baselines is minimal. Perhaps this suggests that the role of activation function encoding for this task is minimal as shown in the original neural functional works and those of [1, 2, 3, 4].

- There is a focus on very specific activation functions (relu, sin, tanh), admittedly, these are common in many architectures, but the authors do not provide any discussion of how the proposed method can be applied to other activation functions.

**Questions:**

- Minimal equivariance is the goal for this task as equivariance is generally easy to achieve. How does the proposed model satisfy minimal equivariance while ignoring permutations that are not functionally equivarant?

- Can the model handle a mixed modelzoo with different activation functions? Note that [1] can handle this case and [3] already demonstrates that this is possible. Per my understanding, the proposed model will need different instantiations of the model for each modelzoo with a different activation function.

- How are the input/output layers of the considered networks handled in the proposed framework? Note that these layers do not follow the permutation symmetries of MLPs(only the hidden layers do).

## References

[1] Lim, Derek, et al. "Graph metanetworks for processing diverse neural architectures." arXiv preprint arXiv:2312.04501 (2023).

[2] Kofinas, Miltiadis, et al. "Graph neural networks for learning equivariant representations of neural networks." arXiv preprint arXiv:2403.12143 (2024).

[3] Andreis, Bruno, Soro Bedionita, and Sung Ju Hwang. "Set-based neural network encoding." arXiv preprint arXiv:2305.16625 (2023).

[4] Unterthiner, Thomas, et al. "Predicting neural network accuracy from weights." arXiv preprint arXiv:2002.11448 (2020).

[5] Zhou, Allan, Chelsea Finn, and James Harrison. "Universal neural functionals." arXiv preprint arXiv:2402.05232 (2024).

**Limitations:**

While limitations are not discussed, i do not see any immediate negative societal impact.

---

> ### Author Rebuttal · Authors · 2024-08-06
>
> Thank you for your thoughtful review and valuable feedback. Below we address your concerns.
>
> **W1: Discussion/comparison with previous works in [C1,C2,C3,C5].**
>
>
> **Answer:** Based on the additional references you provided, we have added the following discussion to the revised version of the paper:
>
> - *Previous methods*: Permutations and scaling (for ReLU networks), as well as sign-flipping (for sine or tanh networks) symmetries, are fundamental symmetries of weight networks. Permutation-equivariant NFNs are successfully built in [C1,C2,C3,C5,33,41,64,65]. In particular, the authors in [C1,C2] carefully construct computational graphs representing the input neural networks' parameters and process the graphs using graph neural networks. In [C3], neural network parameters are efficiently encoded by carefully choosing appropriate set-to-set and set-to-vector functions. The authors in [C5] view network parameters as a special case of a collection of tensors and then construct maximally expressive equivariant linear layers for processing any collection of tensors given a description of their permutation symmetries. These methods are applicable to several types of networks, including those with branches or transformers. However, the models in [C1,C2,C3,C5], as well as others mentioned in our paper, were not necessarily equivariant to scaling nor sign-flipping transformations, which are important symmetries of the input neural networks.
> - *Our method* makes the first step toward incorporating both permutation and non-permutation symmetries into NFNs. In particular, the model proposed in our paper is equivariant to permutations and scaling (for ReLU networks) or sign-flipping (for sine and tanh networks). This leads to a significant reduction in the number of parameters, a property that is particularly useful for large NNs in modern deep learning, while achieving comparable or better results than those in the literature.
>
> **W2: Extensions to architectures with branches/ transformers?**
>
> **Answer:** See **Q2** in General Response.
>
>
> **W3: Not the first to consider activation functions in representing neural network weights.**
>
> **W4: The role of activation function encoding for this task is minimal.**
>
> **Answer to W3-W4:** We are certainly not the first to consider activation functions in representing neural network weights as several previous works in the literature, including [C1,C2,C3,C4], have already done. However, we assert that our models are the first family of NFNs to incorporate non-permutations such as scaling and sign-flipping symmetries of weight spaces, which are crucial symmetries of neural network weights. This leads to a significant reduction in the number of parameters, much lower computational cost and memory consumption, while achieving comparable or better results than those in the literature.
>
> **W5: Applied to other activation functions.**
>
> **Answer:** See **Q1** in General Response.
>
>
> **Q1: How does the proposed model satisfy minimal equivariance while ignoring permutations that are not functionally equivariant?**
>
> **Answer:** By Proposition 4.4, we already excluded false permutations (i.e., permutations that are not functionally equivariant) from the symmetry group $G$ of the weight networks. In addition, by Theorem 5.1, we proved that the minimal equivariance is actually achieved for this group $G$.
>
> **Q2: Can the model handle a mixed modelzoo with different activation functions?**
>
> **Answer:** See **Q2** in General Response. In addition, while [C1,C2,C3] can handle these cases, we do not know whether it is possible to incorporate non-permutations, such as scaling and sign-flipping symmetries which are our main focus, into these models.
>
> **Q3: How are the input/output layers of the considered networks handled in the proposed framework?**
>
> **Answer:** The way the input/output layers are handled is described in Proposition 4.4. According to this proposition, the input and output layers are fixed, as any nontrivial permutation or scaling transformation of the input/output layers would result in a different function.
>
> **References**
>
> [C1] Lim, Derek, et al. Graph metanetworks for processing diverse neural architectures. arXiv 2023.
>
> [C2] Kofinas, Miltiadis, et al. Graph neural networks for learning equivariant representations of neural networks. arXiv 2024.
>
> [C3] Andreis, Bruno, Soro Bedionita, and Sung Ju Hwang. Set-based neural network encoding. arXiv 2023.
>
> [C4] Unterthiner, Thomas, et al. Predicting neural network accuracy from weights. arXiv 2020.
>
> [C5] Zhou, Allan et al. Universal neural functionals. arXiv 2024.

---

> ### Author Response · Authors · 2024-08-10
> **Any Questions from Reviewer RAv9 on Our Rebuttal?**
>
> We would like to thank the reviewer again for your thoughtful reviews and valuable feedback.
>
> We would appreciate it if you could let us know if our responses have addressed your concerns and whether you still have any other questions about our rebuttal.
>
> We would be happy to do any follow-up discussion or address any additional comments.

---

> > ### Author Response · Authors · 2024-08-12
> > **Discussion Deadline is in around Two Days**
> >
> > Dear Reviewer RAv9,
> >
> > We sincerely appreciate the time you have taken to provide feedback on our work, which has helped us greatly improve its clarity, among other attributes. This is a gentle reminder that the Author-Reviewer Discussion period ends in just around two days from this comment, i.e., (11:59 pm AoE on August 13th). We are happy to answer any further questions you may have before then, but we will be unable to respond after that time.
> >
> > If you agree that our responses to your reviews have addressed the concerns you listed, we kindly ask that you consider whether raising your score would more accurately reflect your updated evaluation of our paper. Thank you again for your time and thoughtful comments!
> >
> > Sincerely,
> >
> > Authors

---

### Author Rebuttal · Authors · 2024-08-06

**General Response:**

Dear AC and Reviewers,

Thanks for your thoughtful reviews and valuable comments, which have helped us improve the paper significantly. We are encouraged by the endorsements that: 1) Our Monomial-NFN is a novel contribution to the neural functional literature with sound theoretical results (Reviewer MDVG) and is an important step toward advancing and popularising the field (Reviewer yJwm); 2) Monomial-NFN is equivariant or invariant to a group that has been underexplored (monomial matrices) (Reviewer yJwm); 3) experimental results are compelling (Reviewer MDVG); 4) the parameter saving is a big upside of the Monomial-NFN which can bring better generalization (Reviewer MDVG, RAv9) and be useful for large NNs in modern deep learning (Reviewer yJwm, xQCm).

We address some of the common comments from Reviewers below.

**Q1: Extensibility to other activation functions (Reviewers RAv9, yJwm and xQCm).**

**Answer:** Our approach can be used to handle the symmetries induced by other activation functions, such as LeakyReLU, by using the same strategy based on weight-sharing patterns. Indeed, it is proven in [24] that the linear groups preserved by some other types of activation functions are certain monomial matrices, which is totally similar to our cases.  The reason why we focus on ReLU, sine, and tanh networks is that: they are commonly used in practice, and it is well-known from the literature that the maximal symmetric group of their weight spaces contains both permutations and non-permutations, such as scaling (for ReLU) and sign-flipping (for sine and tanh) symmetries. Both of these symmetries are fundamental symmetries of the weight spaces.

**Q2: Extensibility to other architectures with normalizations/branches/transformers and mixed activations (Reviewers RAv9 and xQCm).**

**Answer:** Our method is applicable to these architectures, provided that the symmetric group of the weight network is known. The idea is to use the weight-sharing mechanism to redefine the constraints of the learnable parameters. The main concern with the architectures that include normalizations/branches/transformers and mixed activation functions is that we do not know whether their weights are equivariant to any non-permutation symmetries, such as scaling or sign-flipping. Since the aim of this paper is to incorporate scaling and sign-flipping symmetries in addition to permutations into NFNs, these architectures are outside the scope of this paper.


**Q3: Compare computation cost and memory to other NFNs (Reviewers MDVG, yJwm and xQCm).**

**Answer:** We have added the runtime and memory consumption of our model and the previous ones in the two tables below to compare the computational and memory costs in the task of predicting CNN generalization. It is observable that our model runs faster and consumes significantly less memory than NP/HNP in [64] and GNN-based method in [B1]. This highlights the benefits of parameter savings in Monomial-NFN.


*Table 1: Runtime of models*
||NP [64]| HNP [64]|GNN [B1]|Monomial-NFN (ours) |
|-|-|-|-|-|
| Tanh subset | 35m34s | 29m37s  |4h25m17s | **18m23s**|
| ReLU subset | 36m40s | 30m06s |4h27m29s | **23m47s**|


*Table 2: Memory consumption*
| | NP [64] | HNP [64]  |GNN [B1] | Monomial-NFN (ours) |
|-|-|-|-|-|
| Tanh subset | 838MB | 856MB |6390MB|**582MB** |
| ReLU subset | 838MB | 856MB |6390MB|**560MB** |


**Q4: Add an example (Reviewers yJwm and xQCm).**

**Answer:**  Let us consider a two-hidden-layers MLP with activation $\sigma=\operatorname{ReLU}$. Assume that $n_0=n_1=n_2=n_3=2$, i.e. all layers have two neurons. This MLP defines a function $f: \mathbb{R}^2 \to \mathbb{R}^2$ given by
$$f(x) = W^{(3)} \sigma \left( W^{(2)} \sigma \left( W^{(1)} x + b^{(1)} \right) + b^{(2)} \right) + b^{(3)},$$
where
$W^{(i)} =\left(W^{(i)}\_{jk}\right)$ is a $2 \times 2$ matrix and $b^{(i)}=[b^{(i)}_1,b^{(i)}_2]^{\top}$ for each $i=1,2,3$. In this case, the weight space $\mathcal{U}$ consists of the tuples $U=(W^{(1)},W^{(2)},W^{(3)},b^{(1)},b^{(2)},b^{(3)})$ and it has dimension 18.

According to Eq. (27), an equivariant layer $E$ over $\mathcal{U}$ has the form $E(U) = (W'^{(1)},W'^{(2)},W'^{(3)},b'^{(1)},b'^{(2)},b'^{(3)})$, where
$$W'^{(1)}\_{jk} = \mathfrak{p}^{1jk}\_{1j1} W^{(1)}\_{j1} + \mathfrak{p}^{1jk}\_{1j2} W^{(1)}\_{j2} + \mathfrak{q}^{1jk}\_{1j} b^{(1)}\_{j},
\qquad \text{and} \qquad b'^{(1)}\_j =  \mathfrak{r}^{1j}\_{1j1} W^{(1)}\_{j1} + \mathfrak{r}^{1j}\_{1j2} W^{(1)}\_{j2} + \mathfrak{s}^{1j}\_{1j}b^{(1)}\_{j},$$
$$W'^{(2)}\_{jk} = \mathfrak{p}^{2jk}\_{2jk} W^{(2)}\_{jk}, \qquad \text{and} \qquad  b'^{(2)}\_j = \mathfrak{s}^{2j}\_{2j}b^{(2)}\_{j},$$
$$W'^{(3)}\_{jk}  = \mathfrak{p}^{3jk}\_{31k} W^{(3)}\_{1k} + \mathfrak{p}^{3jk}\_{32k} W^{(3)}\_{2k}, \qquad \text{and} \qquad  b'^{(3)}\_j =\mathfrak{s}^{3j}\_{31}b^{(3)}\_{1} + \mathfrak{s}^{3j}\_{32}b^{(3)}\_{2} + \mathfrak{t}^{3j}.$$
These equations can be written in a friendly matrix form which we included in the attached pdf. (We move the matrix form to the attached pdf due to space constraint.).

**Reference**

[B1] Graph neural networks for learning equivariant representations of neural networks, ICLR 2024.

---

### Decision · Program_Chairs · 2024-09-25

**Decision:**

Accept (poster)

**Comment:**

This paper generalizes the recently introduced Neural Functional Networks (NFNs) to include also the symmetries caused by particular choices of non-linear activations, i.e., the scaling for ReLU and sign flip for Tanh and Sin. The paper shows that any weight symmetry group of fully connected or convolutional network is in particular a subgroup of their group of consideration, i.e., the “monomial matrix group”. Considering this larger symmetry, the NFNs become more parameter efficient. The paper demonstrates competitive performance to previous NFNs.

The reviewers appreciated the principled approach of the paper and the solid contributions to NFNs. The main limitations raised in the reviews are the worry that these methods suffer from limited expressivity (not improved significantly in the current paper), showing small performance gain over previous work and somewhat limited evaluations. The authors addressed these concerns in rebuttal and offered more experiments and empirical validations including new baselines. The authors also addressed concerns about the extensibility to other activation functions and architectures, and shared some computational and memory cost experiments. Overall the paper seems to offer a strong theoretical advancement and a reasonable empirical advancement to an area of interest and pass the bar for a NeurIPS publication.

We ask the authors to incorporate all the changes and additions discussed in the rebuttal in the camera ready version.